# SLMAP3 is essential for neurulation through mechanisms involving cytoskeletal elements, ABP, and PCP

Taha Rehmani* , Ana Paula Dias* , Billi Dawn Applin , Maysoon Salih, Balwant S Tuana

**SLMAP3 is a tail-anchored membrane protein that targets subcellular organelles and is believed to regulate Hippo signaling. The global loss of SLMAP3 causes late embryonic lethality in mice, with some embryos exhibiting neural tube defects such as craniorachischisis. We show here that SLMAP3$^{-/-}$ embryos display reduced length and increased width of neural plates, signifying arrested convergent extension. The expression of planar cell polarity (PCP) components Dvl2/3 and the activity of the downstream targets ROCK2, cofilin, and JNK1/2 were dysregulated in SLMAP3$^{-/-}$ E12.5 brains. Furthermore, the cytoskeletal proteins (γ-tubulin, actin, and nestin) and apical components (PKCζ and ZO-1) were mislocalized in neural tubes of SLMAP3$^{-/-}$ embryos, with a subsequent decrease in colocalization of PCP proteins (Fzd6 and pDvl2). However, no changes in PCP or cytoskeleton proteins were found in cultured neuroepithelial cells depleted of SLMAP3, suggesting an essential requirement for SLMAP3 for these processes in vivo for neurulation. The loss of SLMAP3 had no impact on Hippo signaling in SLMAP3$^{-/-}$ embryos, brains, and neural tubes. Proteomic analysis revealed SLMAP3 in an interactome with cytoskeletal components, including nestin, tropomyosin 4, intermediate filaments, plectin, the PCP protein SCRIB, and STRIPAK members in embryonic brains. These results reveal a crucial role of SLMAP3 in neural tube development by regulating the cytoskeleton organization and PCP pathway.**

## Introduction

During neurulation, the neural plate organizes into a pseudostratified epithelium that allows this plate to bend and form bilateral neural folds, which fuse to generate the neural tube (1, 2). In this process, apical constriction, driven by contractile forces of actin–myosin on the apical surface, allows neuroepithelial cells to adopt a wedge or bottle shape,

contributing to the bending of the neural plate (3). This remodeling of the cytoskeleton is believed to be critically linked to planar cell polarity (PCP) and apical–basal polarity (ABP), processes that ensure appropriate distribution of proteins and subcellular organelles to maintain tissue polarity (4). Although research is being conducted to unravel the players involved in polarity, the molecular mechanisms that orchestrate the dynamics of these factors, organelle positioning, and cytoskeleton remodeling remain poorly understood. A manifestation of perturbing PCP is neural tube defects (NTDs), a common structural birth abnormality in humans, which can present with lethal consequences (5, 6).

We have previously defined the sarcolemmal membrane–associated protein (SLMAP) as a component of centrosome, ER, mitochondria, and perinuclear membrane, with the potential to impact the function of these organelles (7, 8, 9, 10, 11, 12). SLMAPs are categorized as three main isoforms known as SLMAP1 (~37 kD), SLMAP2 (~45 kD), and SLMAP3 (~83–91 kD), generated by alternative splicing and multiple start sites from a single gene (9, 13). SLMAP is highly conserved, displaying coiled-coil domains along its structure and an alternatively spliced transmembrane domain (TM1/TM2) that targets SLMAP to subcellular membranes (7). The largest isoform SLMAP3 houses an N-terminal forkhead-associated domain (FHA) that binds to phosphothreonines and targets the microtubule-organizing center (MTOC) (11, 14). SLMAP3 is ubiquitously expressed and developmentally regulated, whereas other SLMAPs are tissue-specific (10, 15). Recent studies have indicated that SLMAP3 can negatively regulate the Hippo signaling pathway by interacting with Hippo kinase MST1/2, inducing its dephosphorylation via the striatin-interacting phosphatase and kinase (STRIPAK) complex leading to activation of the transcriptional coactivators YAP/TAZ, which boost proliferation and repress apoptosis (16, 17).

Recently, we reported that the loss of SLMAP3 results in embryonic lethality (12, 18) and our aim here was to discern the role of SLMAP3 specifically in neural tube development. We found no evidence that SLMAP3 impacts Hippo signaling, but rather plays a pivotal role in organizing the cytoskeleton and PCP components in neural tube formation.

---

Department of Cellular and Molecular Medicine, Faculty of Medicine, University of Ottawa, Ottawa, Canada

Correspondence: btuana@uottawa.ca
*Taha Rehmani and Ana Paula Dias are cofirst authors

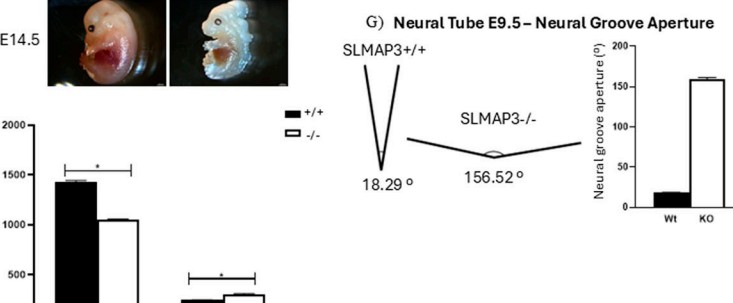

**Figure 1. Loss of SLMAP3 leads to defective neurulation.**
**(A)** Western blot image of E11.5 whole embryo lysates genotyped as WT (+/+), heterozygous (+/−), or KO (−/−) incubated with anti-SLMAP to determine SLMAP isoform expression. Bar graphs represent the expression of SLMAPs by normalizing with total protein visualized by Stain-Free technology. N = 6. *P < 0.05. The P-value was calculated using one-way ANOVA statistical analysis. **(B)** Representative neonatal pups underwent dissection microscopy to magnify the phenotypic differences in animals genotyped as WT (SLMAP3^{+/+}), heterozygous (SLMAP3^{+/−}), or KO (SLMAP3^{−/−}) with a closed and open NT (neural tube). Images were captured with Stereo V.20 to demonstrate size differences, facial retrusion (stunted snout), tail formation, and spine and brain development. Lens magnification = 1x; microscope magnification = 7.5x; scale bar = 0.1 mm. Bar graphs represent length differences, craniofacial retrusion, and tail length in experimental animals (+/−, −/−) compared with the WT. *P < 0.05. The P-value was calculated using one-way ANOVA statistical analysis. N = 3. **(C)** Table displaying the penetrance of embryos displaying neural tube defects because of the

# Results

## Loss of SLMAP3 leads to defective neurulation

To assess the in vivo role of SLMAP3, we have previously generated global SLMAP3 KO mice using the Cre-Lox system ([12]). We followed a breeding strategy to generate cre-independent animals (Fig S1A), because of the risks of DNA damage in mice harboring the recombinase ([19]). LoxP sites were inserted flanking exon 3 of SLMAP (Fig S1B), which specifically nullified the expression of the (91 kD) SLMAP3 isoform but not the expression of SLMAP1 (~37 kD) and SLMAP2 (~45 kD), as indicated in Western blots of E11.5 embryonic lysates (Fig 1A). On postnatal day (P)0, SLMAP3-null pups were stillborn, and all displayed signs of late fetal or neonatal lethality and developmental deformities, as reported previously ([18]) (Fig 1B). $SLMAP3^{-/-}$ embryos were significantly smaller, and displayed facial retrusion (stunted snout) and a truncated tail when compared to $SLMAP3^{+/-}$ or WT littermates (Fig 1B). Furthermore, 9 out of total of 29 SLMAP3-null embryos were also found to exhibit the fatal NTD craniorachischisis (Fig 1C), which presents an open spinal cord and a protruding brain. NTDs are a common birth defect in humans and have been linked to aberrant PCP signaling ([20]).

During early neurulation at E8.5, $SLMAP3^{-/-}$ embryos were significantly shorter (−25%, $P < 0.05$) and the caudal neural plate (posterior neuropore) was significantly wider (31%, $P < 0.05$) than WT embryos (Fig 1D). Because craniorachischisis was the most common NTD phenotype noted in the $SLMAP3^{-/-}$ embryos, transversally sectioned $SLMAP3^{-/-}$ embryos at E9.5 were further examined at the hindbrain–cervical boundary (closure site 1), a site where craniorachischisis defects can occur. Open neural tubes were found to be underdeveloped at E8.5 and E9.5 in $SLMAP3^{-/-}$ when compared to the WT (Fig 1E). Neurulation is regulated by a tissue morphogenic process known as convergent extension, which narrows tissue at the medial–lateral axis while elongating them at the anterior–posterior axis ([21]). To determine whether $SLMAP3^{-/-}$ embryos display convergent extension defects, we analyzed neural plate sections at E8.5 and E9.5 and examined the neural plate thickness and neural grove aperture. The loss of SLMAP3 caused significantly thinner neural plates at E8.5 (−27.42%, $P < 0.05$) and E9.5 (−13.38%, $P < 0.05$) (Fig 1F) and a significantly wider neural grove aperture (772%, $P < 0.05$) in E9.5 neural tubes when compared to WT (Fig 1G). The decreased length and broader width of $SLMAP3^{-/-}$ embryos coupled with thinner and wider neural plates are consistent with convergent extension deficits ([22], [23]). The analysis of our previous RNA-seq ([12]) indicated that target genes commonly associated with neurulation and craniorachischisis were not significantly altered by the loss of SLMAP3 (Fig 1H), implying that SLMAP3 may have a mode of action in neural tube (NT) development at the post-translational level.

## The defects in neurulation caused by SLMAP3 loss are independent of Hippo signaling

Because SLMAP3 was shown to negatively regulate the Hippo signaling pathway via the STRIPAK complex (Fig 2A), we investigated whether SLMAP3 loss could impact Hippo and account for any observed phenotypes. Western blot analysis on E11.5 whole embryo lysates indicated no significant differences in the expression of STRIPAK components (striatin, STRIP1, and PP2A/C) (Fig 2B), or activity in the Hippo-activating kinase MST1/2 or downstream target of Hippo signaling, YAP (Fig 2C). We also assessed E12.5 brains, but no changes in YAP phosphorylation were detected either (Fig 2D), nor did we find changes in YAP localization in E9.5 neural tubes lacking SLMAP3 (Fig S2A). Proliferation assessed with anti-pH3 staining did not suggest changes either (Fig S2B). The reanalysis of our previous RNA-seq data ([12]) (GEO accession number: GSE230748) with heatmaps and Euclidian clustering indicated no changes in Hippo-related gene expression in the E11.5 SLMAP3-deficient embryos (Fig 2E). Together, these results indicate that SLMAP3 is crucial for neurulation by mechanisms that do not involve the Hippo pathway.

## Loss of SLMAP3 perturbs components of PCP

The non-canonical Wnt signaling PCP provides directional cues to tissues in a plane, which affects cell polarity, orientation, and migration, processes essential for convergent extension ([24]). The core PCP proteins consist in two modules that are asymmetrically distributed throughout the cell. These correspond to Vangl1/2 and Prickle proteins on the one side and Frizzled (Fzd) and DVL proteins on the opposite end (Fig 3A). The binding of the Wnt ligand to the Fzd receptor triggers a signaling cascade that modulates ROCK, JNK1/2, and cofilin-1 activities, resulting in actin cytoskeleton rearrangement that leads to the polarization of the tissue (Fig 3A) ([25], [26]). Although the mechanism for the exclusive localization of PCP modules remains elusive, it is appreciated that a competitive interaction between PCP proteins creates the asymmetry ([27]).

Western blot analysis of E12.5 brain lysates from $SLMAP3^{-/-}$ embryos, an organ with only SLMAP3 isoform expression (Fig 3B), indicated a significant change in the protein expression of DVL3 (−56%, $P < 0.05$) and DVL2 (108%, $P < 0.05$) compared with WT, without any changes in DVL1 (Fig 3C). Furthermore, phospho-to-total ratios of the downstream targets of PCP, including ROCK2 (−25, $P < 0.05$), cofilin (40%, $P < 0.05$), and JNK1/2 (−52%, $P < 0.05$), were also significantly altered in SLMAP3-null brain lysates (Fig 3C). We also observed significantly reduced colocalization of Fzd6 and pDVL2, as indicated by the reduced Pearson coefficient in the E8.5 $SLMAP3^{-/-}$ neural tube, corroborating with reduced Wnt signaling activation

---

loss of SLMAP3. NTDs occur in 38% of $SLMAP3^{-/-}$ embryos and consistently present with craniofacial retrusion, growth retardation, and short tail phenotypes. **(D)** Whole-mount images of WT and KO embryos during prenatal development (E8.5-E13.5). Scale bar = 1 mm. The bar graph represents body size and posterior neuropore in WT and KO E8.5 embryos, n = 3. **(E)** Transverse sectioning of WT ($SLMAP^{+/+}$) and KO ($SLMAP^{-/-}$) E8.5 and E9.5 embryos at closure site 1. Sections were stained with hematoxylin and eosin to visualize the cytoplasm and the nucleus. Scale bar = 50 $\mu$m. **(F)** Bar graphs comparing changes in the neural plate thickness in E8.5 and E9.5 neural tubes, n = 3. **(G)** Diagram and bar graph comparing changes in the E9.5 neural grove aperture in WT and KO sections, n = 3. **(H)** Heatmap representing results of RNA sequencing on E11.5 WT and KO whole embryo identifying genes associated with craniorachischisis with Euclidean clustering to highlight similarities in gene expression and experimental animals. The red color signifies lower $\log_2 FC$, whereas green represents higher $\log_2 FC$. N = 4. *$P < 0.05$. The P-value was calculated using an unpaired two-way t test.

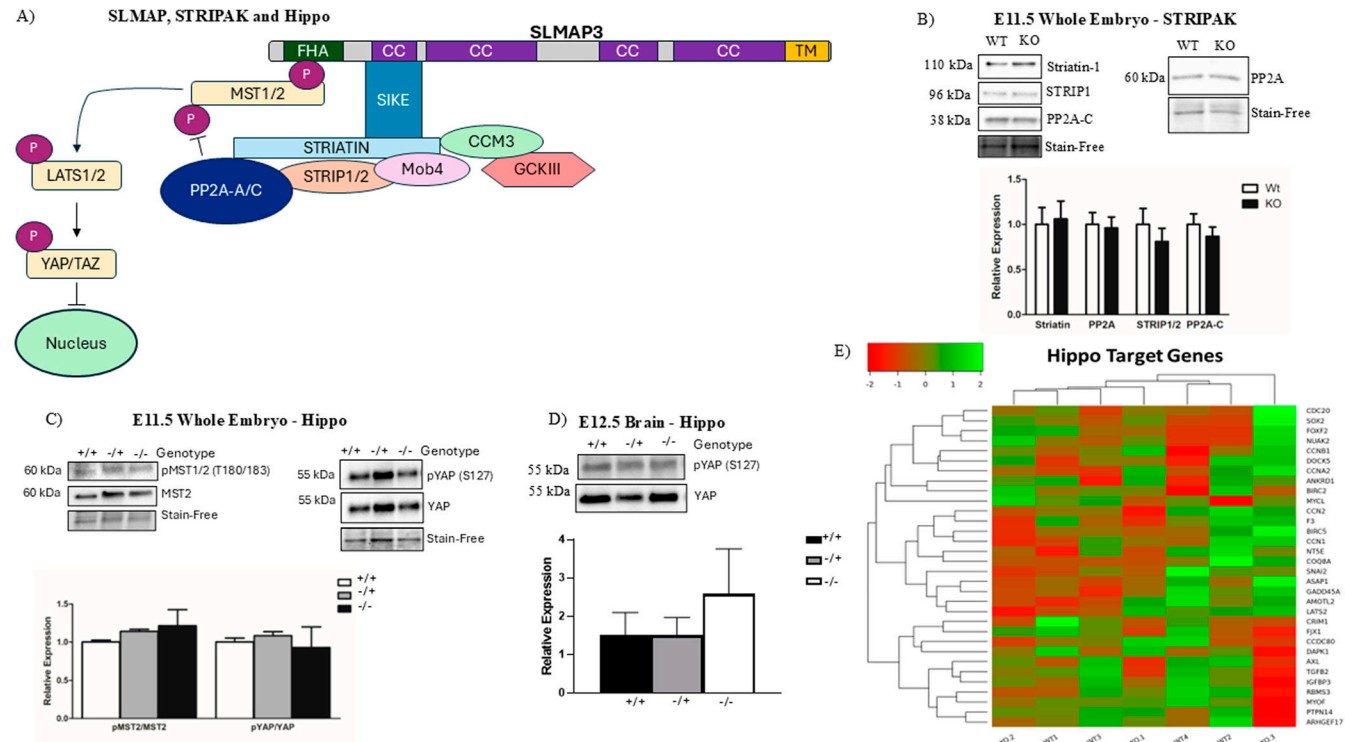

**Figure 2. Defects in neurulation caused by SLMAP3 loss are independent of Hippo signaling.**
**(A)** Proposed schematic of SLMAP-STRIPAK components in Hippo signaling. **(B)** Western blots of *WT* or *KO* lysates from E11.5 whole embryos with anti-STRIPAK components (striatin-1, STRIP1, and PP2A-A/C). N = 3. **(C)** Western blots of *WT* (+/+), heterozygous (−/+), or *KO* (−/−) lysates from E11.5 whole embryos and activity of Hippo-activating kinase, with anti-phospho/total MST2, and downstream transcriptional coactivator, with anti-phospho/total YAP. N = 3. **(D)** Western blots of *WT* (+/+), heterozygous (−/+), or *KO* (−/−) lysates from E12.5 brains and activity of the Hippo downstream transcriptional coactivator, with anti-phospho/total YAP. Bar graphs represent the relative expression or activity of indicated proteins by normalizing to Stain-Free or the phospho-to-total protein ratio. N = 3. **(E)** Heatmap representing results of RNA sequencing on E11.5 *WT* and *KO* whole embryos identifying Hippo signaling target genes with Euclidean clustering to group similarities in gene expression and experimental animals. The red color signifies lower log$_2$FC, whereas green represents higher log$_2$FC. N = 4.

(Fig 3D). However, the expression of other PCP components such as Scribble (SCRIB) or Vangl1/2 was not significantly altered (Fig 3E). To identify whether changes in DVL expression could impact canonical Wnt signaling, we investigated the activity and the expression of *β*-catenin, but no significant alterations were detected (Fig 3F). The data indicate that SLMAP3 regulates the activity of several key components of PCP signaling in brain tissues, which could explain the neurulation defects observed because of its loss.

### Loss of SLMAP3 impacts ABP components

Because the activity of cofilin-1 (actin-severing protein) was dysregulated in the *SLMAP3*$^{−/−}$ brain, we examined whether this could impact actin filament formation at the apical region of the neural tube, as apical constriction of actin in neuroepithelial cells (NEPCs) is necessary to generate forces that bend the developing neural tube (3). For that, we stained E8.5 and E9.5 neural tubes with phalloidin and measured the staining intensity across the neural tube, from the basal to the apical region, with plot profile lines. The distances of these lines and staining intensities were normalized to allow the comparison of multiple neural tubes (Fig 4A). We observed that the neural tubes of *SLMAP3*$^{−/−}$ embryos had significantly

reduced F-actin in the apical region at both E8.5 (Fig 4B) and E9.5 (Fig 4C).

Because apical actin is important for the establishment of ABP in the epithelium (28), we examined whether this process was also affected by analyzing the atypical protein kinase c zeta (PKC*ζ*). This enzyme, which normally localizes to the apical side of epithelial tissues, was found more dispersed across the neural tube of *SLMAP3*$^{−/−}$ embryos (Fig 4D). The tight junction (TJ) protein ZO-1, which is important for ABP (29), was also found less concentrated in the neural tube of E8.5 (Fig 4E) and E9.5 (Fig 4F) *SLMAP3*$^{−/−}$ embryos. Although there was no increase in dispersion across the entire neural tube, ZO-1 appears to be less concentrated and more dispersed within the confined apical region. Given these changes in the apical region, we also assessed the basal regions of the neural tube for nestin, a neurofilament that is normally enriched in the basal region within the neuroepithelium. Nestin was found to be significantly more concentrated in the basal side of E9.5 *SLMAP3*$^{−/−}$ neural tubes (Fig 4G).

Considering the disturbances of ABP in *SLMAP3*$^{−/−}$ neural tubes, we also aimed to investigate adherent junctions (AJs), which are required for this polarity (30). AJs mediate epithelial cell–cell interaction and bind actin filaments at apical ends to aid in apical

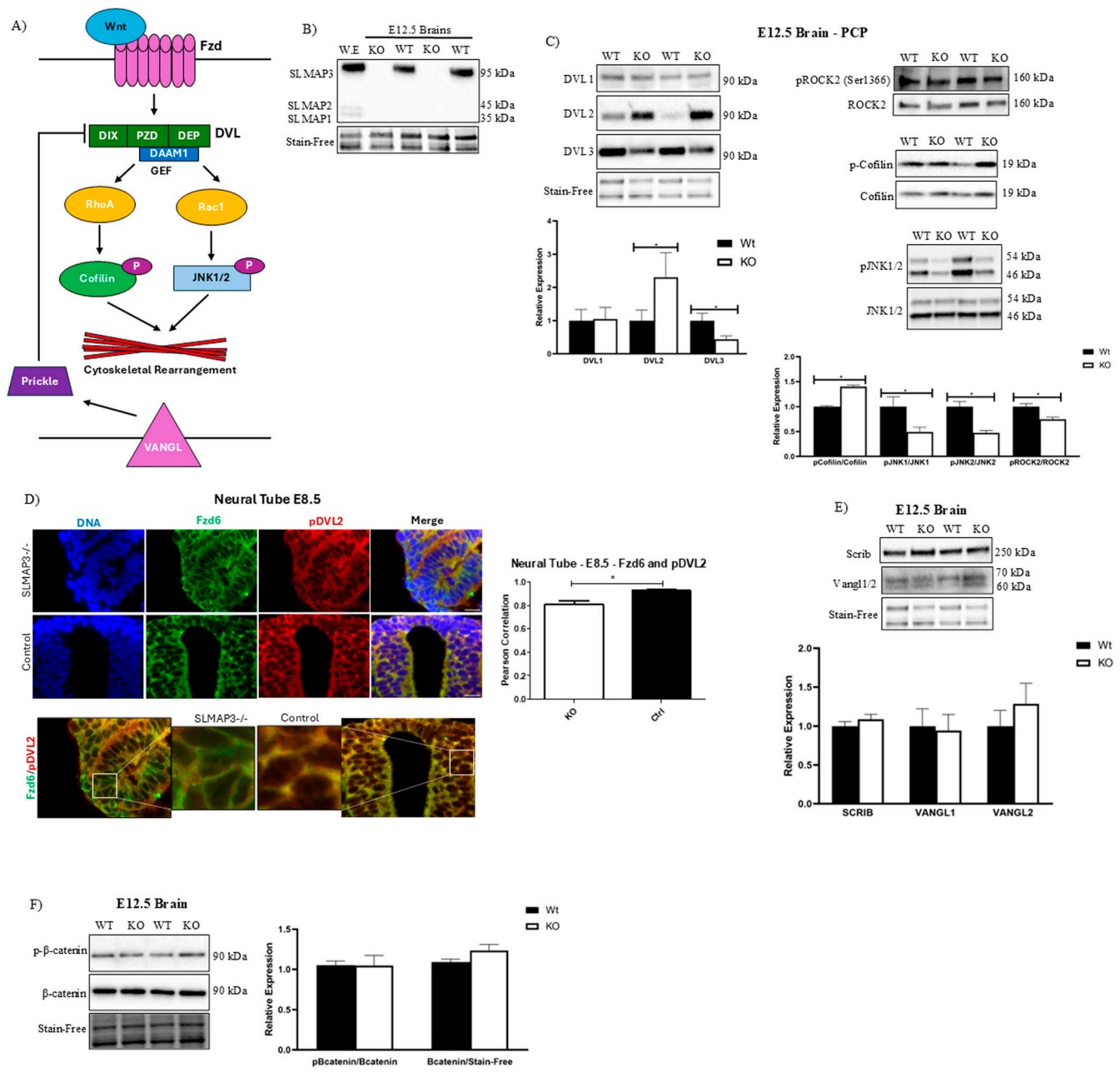

**Figure 3. Loss of SLMAP3 perturbs components of planar cell polarity.**
**(A)** Schematic representing simplified PCP signaling regulating cytoskeletal dynamics through the asymmetrical distribution of Fzd/DVL and Vangl/Prickle. **(B)** Western blot images with anti-SLMAP in *WT* and *KO* E12.5 brain lysates and control whole embryo (W.E) lysates. **(C)** Representative Western blot images with anti-PCP signaling transducers (DVL1, DVL2, DVL3) and PCP signaling targets (ROCK2, cofilin, JNK1/2) in *WT* and *KO* E12.5 brain lysates. Bar graphs represent the relative expression of indicated proteins normalized to Stain-Free total protein or phosphorylated-to-total protein ratio. N = 4. *$P < 0.05$. The *P*-value was calculated using an unpaired two-way *t* test. **(D)** Immunofluorescent imaging of sectioned E8.5 control (*WT* or *KD*) and *SLMAP3⁻ᐟ⁻* neural tubes labeled with DAPI (DNA) (blue), anti-Fzd6 (green), and anti-phospho-DVL2 (red). Scale bar = 20 μm. Bar graphs represent Pearson's correlation of Fzd6 and phospho-DVL2 staining of three images of one neural tube. The *P*-value was calculated using an unpaired two-way *t* test. **(E)** Western blots with anti-SCRIB, anti-Vangl1, and anti- Vangl2 in *WT* and *KO* E12.5 brain lysates. Bar graphs represent the relative expression of indicated proteins normalized to Stain-Free total protein. N = 4. **(F)** Western blots with anti-phospho-*β*-catenin and anti-*β*-catenin in *WT* and *KO* E12.5 brain lysates. Bar graphs represent the relative expression of phospho-to-total B-catenin. N = 4.

constriction, which is necessary for bending epithelial tissues (30). During neurulation, the neural plate expresses varying cadherin types at the different developmental stages (31). At E7.5, the

expression of E-cadherin is notable, and by E8.5, it switches to N-cadherin, which predominates by E9.5 as the neural tube fuses (32). Furthermore, the transition of NEPCs expressing E-cad to N-cad is

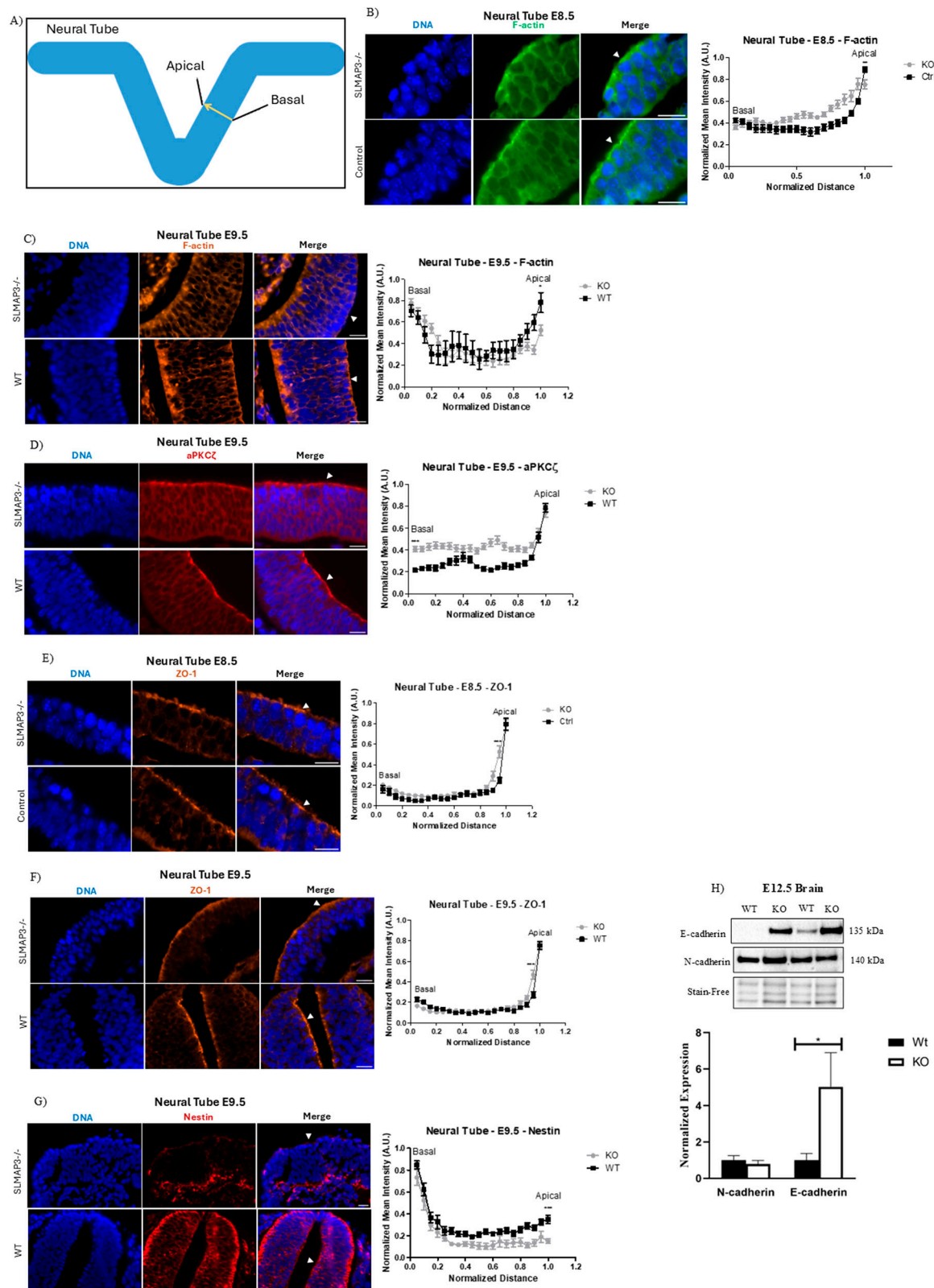

**Figure 4. Loss of SLMAP3 impacts apical basal polarity components.**

**(A)** Schematic representing how analyses in neural tubes were performed. Three lines from the basal to the apical region in each image were drawn for plot profile analysis. Distances and staining intensities were normalized to allow comparison with multiple profilers. **(B)** Immunofluorescent imaging of sectioned E8.5 control and *SLMAP3⁻/⁻* neural tubes labeled with DAPI (DNA) (blue) and phalloidin (F-actin) (green) showing the less concentration of F-actin in *SLMAP3⁻/⁻* neural tubes. Scale bar =

essential for proper neurulation (33). We found that brains of E12.5 *SLMAP3*⁻ᐟ⁻ embryos have the significantly increased expression of E-cadherin (Fig 4H), which could be an indication of dysregulation of adherent junctions. However, we were unable to discern by immunofluorescence if E-cadherin expression in *SLMAP3*⁻ᐟ⁻ neural tubes was increased because of background staining with these antibodies. N-cadherin was equally well expressed in *SLMAP3*⁻ᐟ⁻ and *WT* neural tubes at E8.5 and E9.5, with no changes in distribution across the neural tube (Fig S3A), and no notable changes in the distribution of β-catenin at E8.5 (Fig S3B) and E9.5 (Fig S3C) neural tubes were found either. Therefore, the increased expression of E-cadherin in the brain is most likely due to differentiation abnormalities in the different cell-type constituents rather than mislocalization of AJs in neural tubes. Thus, the lack of SLMAP3 seems to affect the distribution of specific ABP components in neural tubes, exemplified by F-actin, PKCζ and ZO-1, and nestin, but not AJs.

### SLMAP3 associates with cytoskeletal elements

The SLMAP3 interactors determined with immunoprecipitation followed by mass spectrometry (IP-MS) assays of E12.5 brain lysates in our recent study (12) were found to distribute into distinct groups composed of actin regulators, cytoskeleton proteins, neurofilaments, STRIPAK members (34, 35), microtubule motor proteins, and centrosomal, Golgi, and nuclear envelope proteins (Fig 5A). SLMAP3 interactors enriched for actin and intermediate filament processes (Fig 5B), and the interactions with STRN, SCRIB, and nestin were further confirmed here by IP and Western blotting (Fig 5C).

Considering the association of SLMAP3 with the centrosome (11, 12, 36), we assessed γ-tubulin in E9.5 neural tubes and observed that its localization in the apical region was significantly reduced in embryos lacking SLMAP3 (Fig 5D). Interestingly, no changes in localization of SCRIB across the neural tube were observed (Fig 5E), despite the disturbances of ABP because of SLMAP3 loss (Fig 4), and distribution of Vangl2, another PCP member associated with NTDs (37), was not changing either (Fig 5F).

Because we also detected SLMAP3 interaction with Golgb1/Giantin and Golga3 (Fig 5A) and given the association of STRIPAK with Golgi polarization (38), we investigated whether this organelle was abnormal in the *SLMAP3*⁻ᐟ⁻ neural tubes by measuring Golgi vesicle dimensions. However, no changes were observed in the area and major axis length of Golgi particles (Fig S4A), suggesting that SLMAP3 does not affect the organization of this organelle in the neural plate.

In our IP-MS, we found that SLMAP3 interacts with numerous cytoskeletal elements such as actin, microtubule, and intermediate filament cytoskeleton regulators (Fig 5A). These data imply that the cytoskeleton is one of the main targets for SLMAP3 in the developing embryonic brain. Indeed, we detected subtle changes in the dimensions of NEPCs isolated from *SLMAP3*⁻ᐟ⁻ neural plates (Fig 5G), which is in accordance with our previous findings of lack of SLMAP3 impacting the size of cardiomyocytes (39) and mouse embryonic fibroblasts (18). However, Golgi dimensions and vesicle numbers in primary NEPCs showed no changes (Fig S4B) like that observed in neural tubes.

### Knockdown of SLMAP3 in NE-4C cells affects proliferation upon differentiation

Because of the difficulties in generating *SLMAP3*⁻ᐟ⁻ embryos and inconsistent and insufficient isolation of primary NEPCs, we used the epithelial mouse neural stem cell line NE-4C cells to investigate SLMAP3 functions. To this end, we used shRNAs called S4 and S5 that target exon 22 of SLMAP (Fig 6A), and the shRNA control scramble, which we used previously to deplete all SLMAP isoforms (39). NE-4C cells only express SLMAP3, and both shRNA-S4 and shRNA-S5 successfully depleted SLMAP3 by ~90% (Fig 6B). Examination of proliferation of NE-4C cells by staining with anti-pH3 did not show significant changes because of SLMAP3 depletion (Fig 6C). To assess whether SLMAP can impact growth during differentiation, we examined NE-4C cell proliferation with anti-Ki67 staining after 5 d of differentiation with retinoic acid and found a small but significant reduction in proliferation of SLMAP3-depleted NE-4C cells (Fig 6D) without any changes in cell death assayed by TUNEL (Fig 6E). It is notable that depletion of SLMAP reduced the proliferation of both cardiomyoblasts (39) and skeletal myoblasts (12). Thus, SLMAP may influence cell proliferation in a cell/tissue-specific manner.

### SLMAP3 does not impact the distribution of ABP and PCP components in NE-4C cells

Considering the disturbances in ABP observed in the neural tube of *SLMAP3*⁻ᐟ⁻ embryos, we investigated whether the issue stems from defective recruitment of F-actin, TJs (ZO-1), and AJs (N-cadherin) to cell–cell contacts using the NE-4C cell. The measurements of phalloidin staining in NE-4C cells did not show changes in the

---

20 μm. Three lines in each of the multiple figures from one neural tube of each genotype were considered, totalizing n = 12 for *KO* and n = 12 for control. **(C)** Immunofluorescent imaging of sectioned E9.5 *WT* and *SLMAP3*⁻ᐟ⁻ neural tubes labeled with DAPI (DNA) (blue) and phalloidin (F-actin) (orange) also indicating less apical localization of F-actin in *SLMAP3*⁻ᐟ⁻ neural tubes. Scale bar = 20 μm. Three lines in each figure from one neural tube of each genotype were considered, totalizing n = 9 for *KO* and n = 6 for *WT*. **(D)** Immunofluorescent imaging of sectioned E9.5 *WT* and *SLMAP3*⁻ᐟ⁻ neural tubes labeled with DAPI (DNA) (blue) and anti-aPKCζ (red) showing dispersed aPKCζ in *SLMAP3*⁻ᐟ⁻ neural tubes. Scale bar = 20 μm. Three lines in each of the multiple figures from two neural tubes of each genotype were considered, totalizing n = 18 for *KO* and n = 15 for *WT*. **(E)** Immunofluorescent imaging of sectioned E8.5 control (*WT/KD*) and *SLMAP3*⁻ᐟ⁻ neural tubes labeled with DAPI (DNA) (blue) and anti-ZO-1 (orange) showing dispersed ZO-1 in the apical region of *SLMAP3*⁻ᐟ⁻ neural tubes. Scale bar = 20 μm. Three lines in each of the multiple figures from two neural tubes of each genotype were considered, totalizing n = 12 for *KO* and n = 12 for control. **(F)** Immunofluorescent imaging of sectioned E9.5 *WT* and *SLMAP3*⁻ᐟ⁻ neural tubes labeled with DAPI (DNA) (blue) and anti-ZO-1 (orange) showing dispersed ZO-1 in the apical region of *SLMAP*⁻ᐟ⁻ neural tubes. Scale bar = 20 μm. Three lines in each of the multiple figures from three neural tubes of each genotype were considered, totalizing n = 24 for *KO* and n = 24 for *WT*. **(G)** Immunofluorescent imaging of sectioned E9.5 *WT* and *SLMAP3*⁻ᐟ⁻ neural tubes labeled with DAPI (DNA) (blue) and anti-nestin (red) indicating that nestin is more concentrated in the basal region of *SLMAP3*⁻ᐟ⁻ neural tubes compared with the *WT*, where nestin is also visible across the neural tube. Scale bar = 20 μm. Three lines in each figure from four neural tubes of each genotype were considered, totalizing n = 12 for *KO* and n = 15 for *WT*. **(H)** Western blot images with anti-E-cad and anti-N-cad in *WT* and *KO* E12.5 brain lysates. Bar graphs represent the relative expression of indicated proteins normalized to Stain-Free total protein. N = 4. *P < 0.05, **P < 0.01, and ***P < 0.001. The *P*-value was calculated using an unpaired two-way *t* test. Arrowheads indicate the apical region of the neural tube.

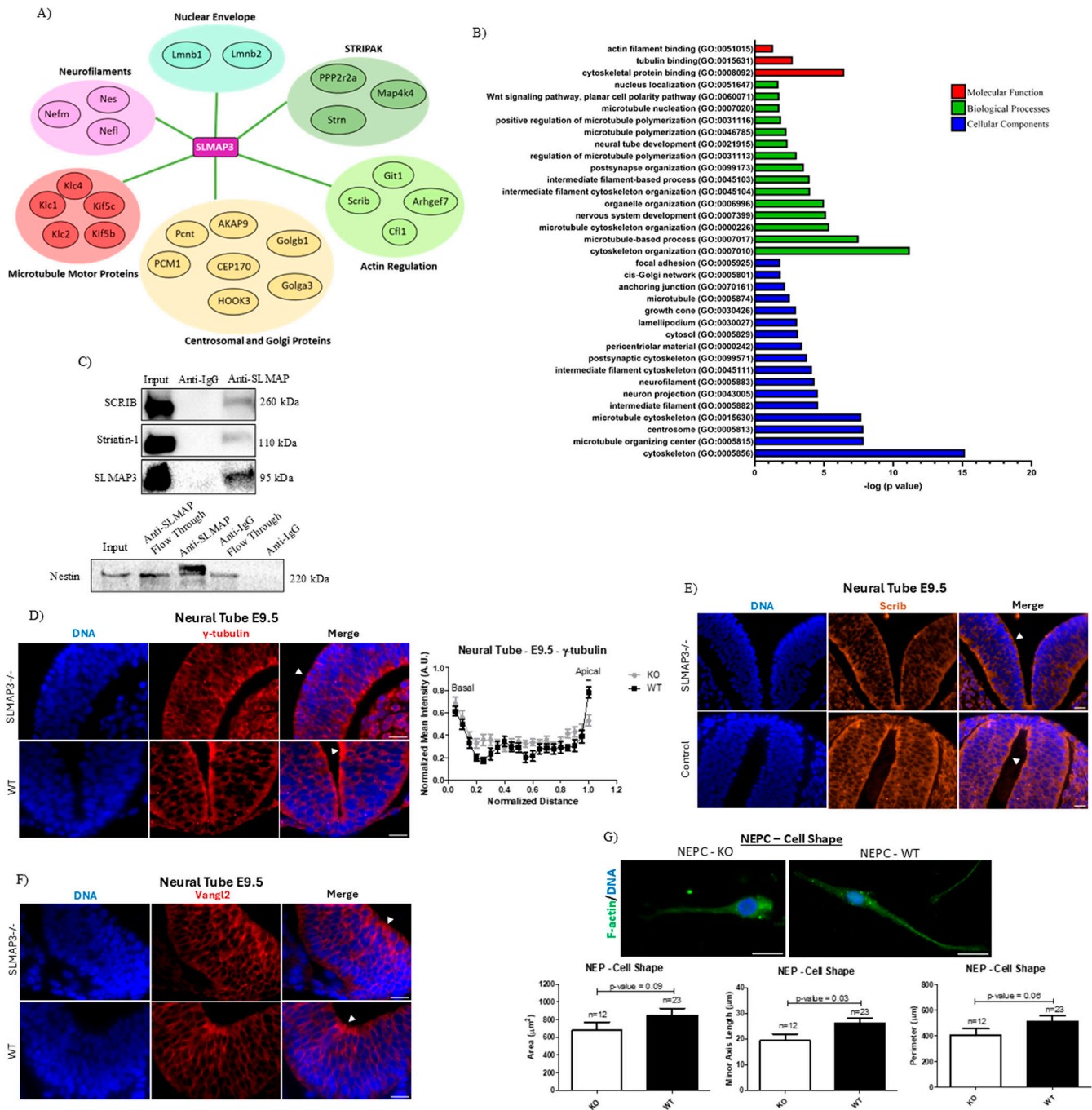

**Figure 5. SLMAP associates with cytoskeletal elements.**
**(A)** Protein–protein interaction network of the highest peptide discovery count for SLMAP3 determined from mass spectroscopy peptide analysis of anti-SLMAP immunoprecipitated E12.5 brain lysates (12). **(B)** GO analysis of molecular function, biological process, and cellular components associated with the network of proteins that interact with SLMAP3. **(C)** Western blot displaying the presence of SLMAP3, striatin-1, SCRIB, and nestin after immunoprecipitation of E12.5 brain lysates with anti-SLMAP and anti-IgG and blotting with anti-SCRIB, anti-striatin-1, and anti-nestin. **(D)** Immunofluorescent imaging of sectioned E9.5 *WT* and *SLMAP3$^{-/-}$* neural tubes labeled with DAPI (DNA) (blue) and anti-γ-tubulin (red), indicating less tubulin in the apical regions of *SLMAP3$^{-/-}$* neural tubes. Scale bar = 20 μm. Three lines in each of the multiple figures from two neural tubes of each genotype were considered, totalizing n = 12 for *KO* and n = 12 for *WT*. **(E)** Immunofluorescent imaging of sectioned E9.5 control (*WT*/*KD*) and *SLMAP3$^{-/-}$* neural tubes labeled with DAPI (DNA) (blue) and anti-SCRIB (orange). Scale bar = 20 μm. **(F)** Immunofluorescent imaging of sectioned E9.5 *WT* and *SLMAP3$^{-/-}$* neural tubes labeled with DAPI (DNA) (blue) and anti-Vangl2 (red). Scale bar = 20 μm. **(G)** Immunofluorescent imaging of *WT* and *KO* primary NEPCs labeled with DAPI (DNA) (blue) and phalloidin (F-actin) (green). Bar graphs represent the area, minor axis length, and perimeter of NEPCs. Scale bar = 8 μm. The number of cells analyzed is indicated in the graphs. **P < 0.01. The *P*-value was calculated using an unpaired two-way *t* test. Arrowheads indicate the apical region of the neural tube.

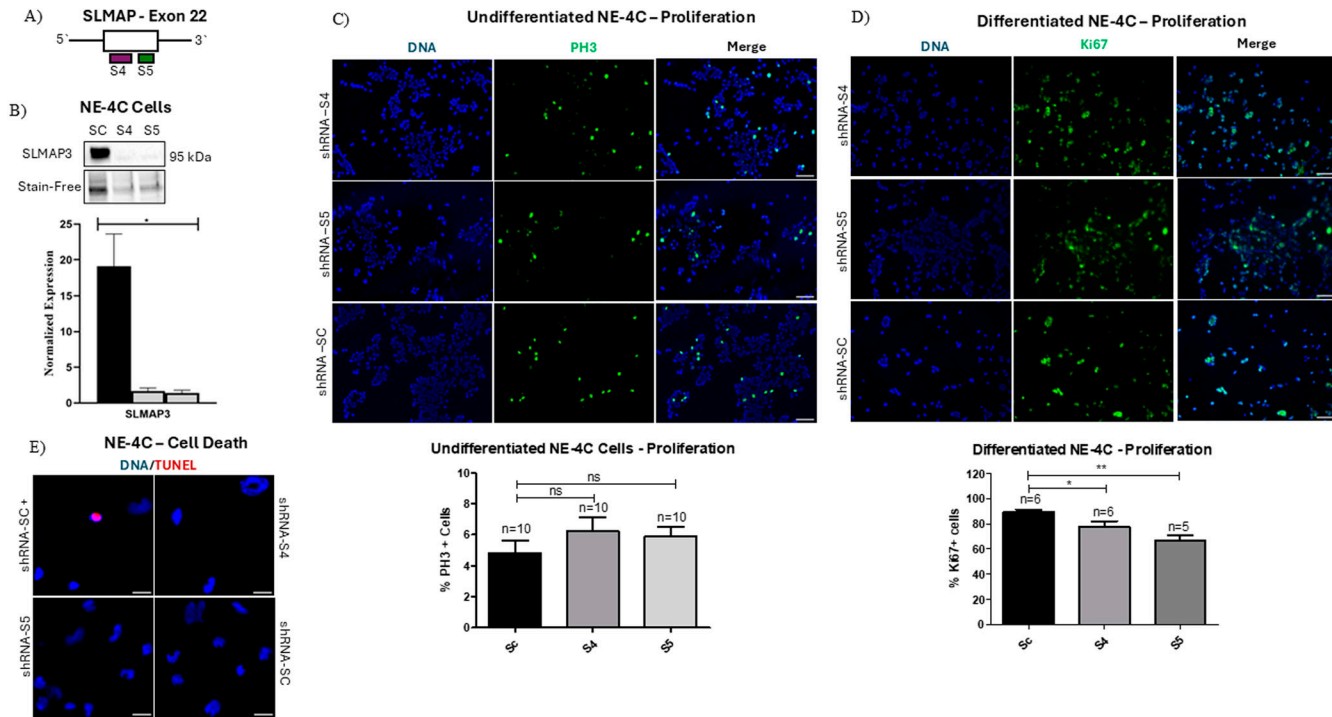

**Figure 6. Knockdown of SLMAP3 in NE-4C cells affects proliferation upon differentiation.**
**(A)** shRNA (S4 & S5) that targets exon 22 of the SLMAP. **(B)** Western blot with anti-SLMAP in NE-4C cell lysates transduced with shRNA-S4, shRNA-S5, and shRNA-Sc. The bar graph represents the quantification of the relative expression of SLMAP3 normalized to Stain-Free total protein. N = 4. **(C)** Immunofluorescent imaging of shRNA-Sc, shRNA-S4, and shRNA-S5 undifferentiated NE-4C cells labeled with DAPI (DNA) (blue) and anti-phospho-histone 3 (pH3) (green). Scale bar = 100 $\mu$m. The number of fields analyzed is indicated in the graphs. Bar graphs represent the percentages of pH3-positive cells. **(D)** Immunofluorescent imaging of shRNA-Sc, shRNA-S4, and shRNA-S5 differentiated NE-4C cells labeled with DAPI (DNA) (blue) and anti-KI67 (green). Scale bar = 100 $\mu$m. The number of fields analyzed is indicated in the graphs. Bar graphs represent the percentages of Ki67-positive cells. **(E)** Immunofluorescent imaging of shRNA-Sc, shRNA-S4, and shRNA-S5 differentiated NE-4C cells labeled with DAPI (DNA) (blue) and TUNEL (red). Scale bar = 20 $\mu$m. *$P < 0.05$ and **$P < 0.01$. The $P$-value was calculated using one-way ANOVA statistical analysis.

accumulation of F-actin in cell junctions with depletion of SLMAP3 (Fig 7A), no alterations were observed for ZO-1 (Fig 7B) or N-cadherin (Fig S5A), and induction of the differentiation of NE-4C cells did affect the N-cadherin recruitment to cell edges and contacts either (Fig S5B). We also assessed the localization of PCP components, considering the alterations observed in $SLMAP3^{-/-}$ neural tubes. SLMAP3-depleted NE-4C cells could still recruit SCRIB (Fig 7C) and Vangl2 (Fig 7D) to cell–cell contacts. Fzd6 could also be recruited, and no differences in its colocalization with pDVL2 were detected (Fig 7E). These results suggest that SLMAP3 is not directly required to recruit ABP and PCP components to cell–cell junctions in the NE-4C cell culture, distinct from the alterations observed in $SLMAP3^{-/-}$ neural tubes in vivo.

### Knockdown of SLMAP3 in NE-4C cells does not affect microtubule nucleation activity

Given the association of SLMAP3 with the centrosomes (11, 12, 36), we sought to investigate any changes in this organelle in NE-4C cells. We did not find any obvious differences in the centrosome or Golgi, as indicated by pericentrin and GM130 staining in SLMAP3-depleted NE-4C cells (Fig S6A). Furthermore, induction of differentiation of NE-4C cells resulted in a similar dispersal of pericentrin in both S4 and SC NE-4C cells (Fig S6B). Then, we conducted

microtubule nucleation assays by inducing microtubule depolymerization with nocodazole, followed by washout and microtubule growth (40), but no changes in the MTOC could be detected in either undifferentiated (Fig S6C) or differentiated (Fig S6D) NE-4C cells. These data are in accordance with our recent findings in MEFs and C2C12, where SLMAP3 was not required for microtubule nucleation (12).

## Discussion

Our data here define a critical role of SLMAP3 in neural tube development, because its loss causes some embryos to exhibit NTDs, including incomplete closure and craniorachischisis. Our data show that the loss of SLMAP3 in vivo resulted in changes in organization of cytoskeletal components such as γ-tubulin, actin, and nestin, which influence subcellular architecture and distribution of proteins that guide ABP and PCP signaling. NT formation is critically dependent on the cytoskeleton, and proper PCP and ABP as a perturbation in these processes cause NTDs (41, 42).

The embryonic KO of the SLMAP3 isoform results in embryonic lethality, with stunted growth characteristics and organ abnormalities, as we reported recently (12, 18). The NT and brain express only the SLMAP3 isoform and thus were severely compromised

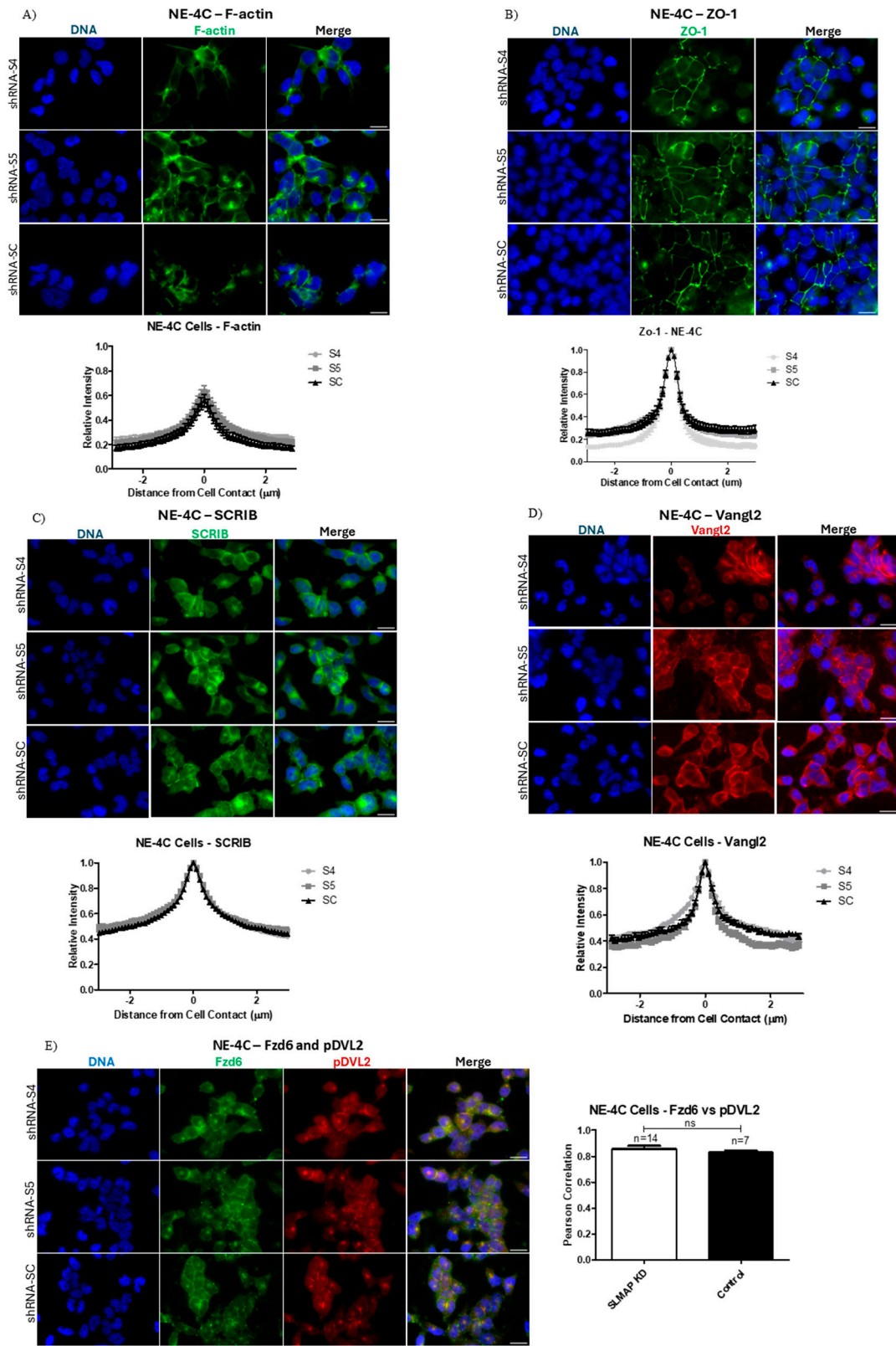

**Figure 7. SLMAP3 does not impact the distribution of ABP and PCP components in NE-4C cells.**
**(A)** Immunofluorescent imaging of shRNA-Sc, shRNA-S4, and shRNA-S5 NE-4C cells labeled with DAPI (DNA) (blue) and phalloidin (F-actin) (green). The graph represents the relative intensity of F-actin in 27 cell–cell contacts across three biological samples. Scale bar = 20 μm. **(B)** Immunofluorescent imaging of shRNA-Sc, shRNA-S4, and shRNA-S5 NE-4C cells labeled with DAPI (DNA) (blue) and anti-ZO-1 (green). The graph represents the relative intensity of ZO-1 in 15 cell–cell contacts. Scale bar = 20 μm.

during development. In this regard, we did not observe any defects in cardiac development in the conditional *KO* of the SLMAP3 isoform in the postnatal mouse heart (39, 43), where the abundant expression of other SLMAP isoforms potentially compensated for the loss of SLMAP3 (39). Although SLMAP3 has recently been linked to Hippo signaling (16, 17), we did not find any impact on this pathway at the phenotypic or molecular level in the brain and in embryos or tissues lacking the SLMAP3 isoform (12, 18). It appears that SLMAP3 mediates organogenesis in a cell- and tissue-specific manner as we recently reported its roles at MTOC in myogenesis (12) and primary cilia in mouse embryonic fibroblasts (18).

NTDs represent some of the most common birth defects globally, approximately affecting 1:1,000 births (2, 44). Many of these defects are linked to folate metabolism and genetic factors; however, a significant number of cases remain idiopathic, particularly craniorachischisis (45). Most observed cases of NTD in *SLMAP3*$^{-/-}$ mice manifested as craniorachischisis, whereas the occurrence of spina bifida and exencephaly was also evident in some embryos. Because SLMAP3 loss leads to these severe NT deficits, it may be worthwhile interrogating human samples of NTDs for any links to SLMAP.

The phenotypes observed with the loss of SLMAP are consistent with those observed with ablation or suppression of the PCP component (46, 47, 48). The PCP signaling pathway is essential for convergent extension, which is clearly aberrant at E8.5 and E9.5, as exemplified by the reduced length-to-width ratios and significantly wider neural plates in *SLMAP3*$^{-/-}$ embryos. The embryonic loss of *SLMAP3*$^{-/-}$ revealed substantial changes in the expression of components of PCP signaling, including DVL2 and DVL3, as well as the downstream targets such as ROCK2, cofilin, and JNK, which have all been associated with NTDs (49). In the absence of SLMAP3 in the neural tubes, the colocalization of Fzd6 and Dvl2, proteins involved in Wnt signaling, was also reduced. The loss of DVL2 and DVL3 has been reported to result in incomplete neural tube closure in mice, whereas mutations in DVL2 have been linked to human NTDs (50). Thus, the dysregulation of DVL2/3 in the neural tube because of SLMAP3 loss may contribute to the observed NTDs, which is in line with what we recently reported in *SLMAP3*$^{-/-}$ embryos (18). SLMAP3-mediated regulation in the DVL2/3 protein levels most likely occurs through post-translational mechanisms (51) as we reported the protection of the DVL3 protein from lysosomal degradation by SLMAP3 (18).

The dysregulation of cofilin and ROCK because of SLMAP3 loss may have altered the actin distribution, which is pivotal in facilitating apical constriction, essential for convergent extension and neurulation (52, 53, 54). In addition, our recent IP-MS revealed SLMAP3 in association with cytoskeletal elements including tropomyosin-4 and linkers such as plectin, which are involved in cytoskeleton regulation (12). These data are consistent with our previous findings in cardiac tissues, where α and β subunits of the myosin heavy chain were associated with SLMAP and colocalized with it at the contractile apparatus (10). The association of SLMAP with contractile machinery suggests that SLMAP may play a general role in regulation of contractile activity in muscle and non-muscle tissues. Because PCP controls actin cytoskeleton dynamics, SLMAP might be involved in linking these processes.

The ZO family of proteins are essential for assembling TJs and are known to connect with the cytoskeleton, including actin (55, 56). The changes in ZO1 may be linked to the altered actin distribution in neural tubes with SLMAP3 deficiency. Furthermore, changes in the apical localization of ZO-1 and PKCζ suggest an impact on ABP. Thus, the defective NT could result from an aberrant ABP/PCP signaling because of SLMAP3 loss. It is notable that we found SCRIB to interact with SLMAP3, although its distribution or expression and that of Vangl2 were not affected in NTs deficient in SLMAP3. SCRIB has been reported to link ABP/PCP and found to regulate the localization and stability of Vangl2, AJs (E-cad/N-cad), and TJs (57, 58, 59, 60). In addition, mutations in SCRIB or its deficiency resemble the phenotypes that we report here with the loss of SLMAP3 (59).

The exploration in NE-4C cells with SLMAP3 knockdown did not recapitulate the data seen with SLMAP3 loss in vivo, but this is not surprising because cell polarity is best studied in organotypic cultures, with controlled extracellular microenvironment to provide polarity cues (61). Therefore, with the findings from this study, we propose the following model: SLMAP3, at the level of cytoskeletal elements, regulates the transduction of PCP signaling through mechanisms involving protein stability of DVLs (18) and interaction with SCRIB, which in turn impacts ABP/PCP. This signaling module containing SLMAP3 then governs cell polarity, and consequently the localization of aPKCζ and the cytoskeleton components γ-tubulin, F-actin, nestin, and ZO-1 in the neural tube. The ability of SLMAP3 to regulate the position of these components in vivo suggests that it may serve roles in the three-dimensional organization and signaling cues that guide PCP. The findings that depletion of SLMAP3 in NE-4C does not disrupt the cellular localization of F-actin and ZO-1 or SCRIB, Vangl2, Fzd6, and pDVL2 further support this view. Therefore, the processing of polarity cues for neural tube development was defective in *SLMAP3*$^{-/-}$ embryos, whereas in the cell culture, these proteins could still be recruited to cell–cell contacts. SLMAP3 can assemble with the STRIPAK in the embryonic brain, and whether that signaling complex can also influence cell polarity remains to be determined. Furthermore, despite the implied interactions with Golgi components, we found no changes in its distribution, whereas the MTOC localization was affected in NT null for SLMAP3. Although the activity of these organelles has been linked to PCP, how and whether SLMAP may influence them remains to be fully interrogated.

Colletively, our data show that SLMAP3 is necessary for neural tube development by regulating the cytoskeleton and PCP

**(C)** Immunofluorescent imaging of shRNA-Sc, shRNA-S4, and shRNA-S5 NE-4C cells labeled with DAPI (DNA) (blue) and anti-SCRIB (green). The graph represents the distribution of the relative intensity of SCRIB in 35 cell–cell contacts across two biological samples. Scale bar = 20 μm. **(D)** Immunofluorescent imaging of shRNA-Sc, shRNA-S4, and shRNA-S5 NE-4C cells labeled with DAPI (DNA) (blue) and anti-Vangl2 (red). The graph represents the distribution of the relative intensity of Vangl2 in 30 cell–cell contacts across two biological samples. Scale bar = 20 μm. **(E)** Immunofluorescent imaging of shRNA-Sc, shRNA-S4, and shRNA-S5 NE-4C cells labeled with DAPI (DNA) (blue), anti-Fzd6 (green), and anti-phospho-DVL2 (red). Scale bar = 20 μm. The bar graph represents Pearson's correlation of Fzd6 and phospho-DVL2 in the SLMAP *KD* (shRNA-S4 and shRNA-S5) and control (shRNA-Sc) neural tube. The number of fields quantified is indicated in the graph. The *P*-value was calculated using an unpaired two-way *t* test.

 **Life Science Alliance**

components to influence convergent extension and neurulation, independent of any impact on Hippo signaling.

# Materials and Methods

### Generating and genotyping the SLMAP3 KO mice

The procedure to create flox-SLMAP mice and to generate global KOs was described previously (12, 43). We crossed flox-SLMAP mice with CMV-cre mice to create a heterozygous flox-SLMAP mouse, which we termed "cre-dependent" breeding. Then, we crossed $SLMAP^{+/−}$ and $WT$ mice to pass on the cleaved SLMAP gene but not cre. In the "cre-independent" breeding, heterozygous mice were crossed to generate $SLMAP3^{−/−}$ or KO animals (Fig S1A). These mice were genotyped by extracting genomic DNA from ear clips by boiling for 10 min at 95°C in 180 $\mu$l of 50 mM NaOH per ear. The DNA solution was then neutralized using 20 $\mu$l of 1 M Tris-Cl, pH 8.0. For genotyping, we used PCR DreamTaq Green PCR Master Mix 2x (Thermo Fisher Scientific). The following primers were used: forward F2 primer (5′-CCTGGAGAGCCTCCGTGTGAGT-3′), reverse R2 primer (5′-GTCAACTGCCCAATGTACAGAAATAGTAAG-3′), and reverse R1 primer (5′-GGAGAGACTATCACAGCCACAGGA-3′) (Fig S1B). Primers F2/R2 will identify LoxP1 sites, the absence of which will suggest cleaved SLMAP3. This is confirmed by primers F2/R1, as the cleaved $slmap$ gene will produce an amplicon size at 400 bp, whereas $WT$-SLMAP will produce an amplicon size of 1,400 bp (Fig S1B). PCR amplicons were visualized by RedSafe (Sigma-Aldrich) staining of a 1% agarose gel after electrophoresis.

Pregnant mothers were plug-checked, and the morning of the first plug was designated as E0.5. Mothers were anesthetized, and embryos were dissected away at the desired stage in ice-cold 1x PBS. Either embryos were fixed in 10% neutral buffered formalin or tissues (brain, whole embryo) were snap-frozen for protein or transcript analysis and/or cultured for cell analysis. Mice were handled in accordance with the guidelines set by the Canadian Council on Animal Care, Guide to the Care and Use of Experimental Animals, 2 vols. (Ottawa, Ont.: CCAC, 1980–1993), and Animals for Research Act, R.S.O. 1990, c.A. 22. All animal protocols and procedures were approved by the Animal Care Committee of the University of Ottawa.

### Histological and immunofluorescent analyses

We paraffin-embedded the fixed embryos and transversally sectioned them targeting the hindbrain–cervical boundary to examine the neural tube closure at this site. Paraffin-embedded slides were processed through well-established protocols for rehydration and citrate buffer antigen retrieval (62). Sectioned slides were stained with hematoxylin and eosin to visualize the neural tube and neighboring tissues.

For immunofluorescence of histological sections, rehydrated slides were subjected to 1 h of 5% BSA in PBS-T (1x PBS, 0.01% Triton X-100) of blocking. After rinsing with PBS, slides were incubated overnight at 4°C with diluted antibodies according to Table 1 in blocking solution. Slides were then washed 5x for 5 min with PBS

and then secondary Alexa 555 (Thermo Fisher Scientific) or Alexa 647 (Thermo Fisher Scientific) and incubated for 1 h at RT. For F-actin staining, phalloidin CF 488 or 555 (Biotium) was incubated together with the secondary antibody and with a 1:40 dilution. We used VectaShield Plus DAPI (MJS Biolynx) mounting medium with 1.5-mm coverslips (Thermo Fisher Scientific) sealed with nail polish and imaged when completely dried.

For immunofluorescent imaging of cells, we placed autoclaved circular 1.5-mm coverslips (VWR) in 12-well plates and incubated with poly-L-lysine (MilliporeSigma) according to the manufacturer's protocol. The poly-L-lysine solution was rinsed with 1x PBS and air-dried in a hood for 30 min before cells were plated onto the wells containing the coverslips. After attachment, cells were rinsed gently with 1x PBS before adding 4% PFA for 20 min. After washing 3 times for 5 min with 1x PBS, we blocked and permeabilized cells for 20 min with PBS-T (1% BSA, 1x PBS, and 0.3% Triton X-100). After three more washes for 5 min each with 1x PBS, we added primary antibody overnight at 4°C. Samples were thoroughly washed five times for 5 min with 1x PBS, and then, we added either secondary Alexa 555 (Thermo Fisher Scientific) or Alexa 647 (Thermo Fisher Scientific) for 1 h at RT. We used VectaShield Plus DAPI (MJS Biolynx) mounting medium on coverslips, placing them to face down on slides, and sealed them with nail polish. Images from the dried slides were captured on Zeiss Axio Imager M2 and analyzed using ImageJ and CellProfiler.

To analyze staining distribution across the neural tube, we used the software Fiji. We drew three individual lines from the basal to the apical region of the neural tube for each image. Plot profile analysis was then applied for the quantification of the intensities across the tissue. Both the distances and the staining intensity were normalized relative to the maximum distance and intensity, respectively, to allow the comparison of the protein distribution across the neural tube, and the results of multiple lines were plotted in the graphs together. To quantify Golgi dimensions, we used the software CellProfiler. For neural tubes, we manually masked the images to allow the quantification only in the neural tubes, whereas for cells, we used the entire image field. We performed background subtraction and image threshold, which was posteriorly converted into objects representing Golgi vesicles/particles. The parameters axis length, area, and perimeter were measured in these objects. For the quantification of staining between cell–cell contacts, we drew at least three lines in these contacts in each image for plot profile analysis on Fiji. In these profiles, we considered the highest signal to be the region of cell–cell contact, which is being distance 0. The distance of the other datapoints was then considered in reference to this point zero. The intensities were also normalized relative to the maximum intensities. The adjusted profiles were then plotted together. For the colocalization analysis, we used the JACoP plugin from Fiji with the Costes threshold, which provides the Pearson correlation coefficient (r) (63). For the quantification of microtubule nucleation in undifferentiated NE-4C cells, we used CellProfiler to identify the sites of microtubule nucleation, indicated by anti-$\alpha$-tubulin staining. These sites were turned into objects, and the staining intensity and area were plotted. For differentiated NE-4C cells, because the sites of microtubule nucleation were dispersed and not clear with the anti-$\alpha$-tubulin staining, we used the anti-

**Table 1.  List of antibodies used in this study.**

| Antibody | Manufacturer | Dilution WB | Dilution IF |
|---|---|---|---|
| SLMAP | NBP1-81397; Novus Biologicals | 1:1,000 | |
| Striatin-1 | 610838; BD Transduction Laboratories | 1:1,000 | |
| STRIP1 | A304-644a-m; Bethyl Laboratories | 1:1,000 | |
| PP2A-A α subunit | 07-250; MilliporeSigma | 1:600 | |
| PP2A-C α/β subunit | sc-80665; Santa Cruz Biotechnology | 1:1,000 | |
| Phospho-MST1/2 (T183/180) | 49332S; Cell Signaling Technology | 1:1,000 | |
| KRS-1 (MST2) | sc-130405; Santa Cruz Biotechnology | 1:1,000 | |
| Phospho-YAP (S127) | 13008S; Cell Signaling Technology | 1:1,000 | |
| YAP | 14074S; Cell Signaling Technology | 1:1,000 | 1:200 |
| Phospho-histone 3 (S10) | ab5176; Abcam | | 1:200 |
| DVL1 | 27384-1-AP; ProteinTech | 1:1,000 | |
| DVL2 | ab22616; Abcam | 1:1,000 | |
| DVL3 | CS 3218; Cell Signaling | 1:1,000 | |
| Phospho-ROCK2 (S1366) | GTX122651; GeneTex | 1:1,000 | |
| ROCK2 | sc-398519; Santa Cruz Biotechnology | 1:1,000 | |
| Phospho-cofilin (S3) | 3313T; Cell Signaling | 1:1,000 | |
| Cofilin | 5175; Cell Signaling | 1:1,000 | |
| Phospho-JNK1/2 (T183/Tyr185) | 9251; Cell Signaling | 1:1,000 | |
| JNK1/2 | 9252; Cell Signaling | 1:1,000 | |
| Fzd6 (C-12) | sc-393791; Santa Cruz Biotechnology | | 1:250 |
| Phospho-DVL2 (S143) | ab124933; Abcam | | 1:200 |
| SCRIB | 27083-1-AP; ProteinTech | 1:1,000 | 1:200 |
| Vangl1 | 14783; Cell Signaling | 1:1,000 | |
| Vangl2 | 21492-1-AP; ProteinTech | 1:1,000 | |
| Phospho-beta-catenin (BC-22) | sc-57535; Santa Cruz Biotechnology | 1:1,000 | |
| beta-catenin (E-5) | sc-7963; Santa Cruz Biotechnology | 1:1,000 | 1:250 |
| PKC zeta | sc-17781; Santa Cruz Biotechnology | | 1:250 |
| ZO-1 | 21773-1-AP; ProteinTech | | 1:200 |
| Nestin | MAB353; MilliporeSigma | 1:1,000 | 1:150 |
| N-cadherin | 22018-1-AP; ProteinTech | 1:1,000 | 1:200 |
| Gamma-tubulin (GTU-88) | T6557; MilliporeSigma | | 1:500 |
| Vangl2 (clone 2G4) | MABN750; MilliporeSigma | | 1:200 |
| GM130 | 610822; BD Biosciences | | 1:200 |
| Ki67 | ab15580; Abcam | | 1:200 |
| Pericentrin | PRB-432C; Covance/BioLegend | | 1:100 |
| Alpha-tubulin | ab176560; Abcam | | 1:100 |
| Alexa 488 | Rabbit Secondary IgG Fluorescent Marker | | 1:300 |
| Alexa 555 | Rabbit Secondary IgG Fluorescent Marker | | 1:300 |
| Alexa 647 | Mouse Secondary IgG Fluorescent Marker | | 1:300 |
| Alexa 647 | Rat Secondary IgG Fluorescent Marker | | 1:300 |
| Anti-rabbit HRP | Jackson ImmunoResearch | 1:10,000 | |
| Anti-mouse HRP | Jackson ImmunoResearch | 1:10,000 | |

All antibodies used in this study are listed with the corresponding distributor, catalog number, and dilution used for Western blot.

pericentrin staining to create objects and measure the intensity of anti-$\alpha$-tubulin staining inside them. For all these analyses, in some cases we considered heterozygous mice, which do not present any phenotype, as controls because of sample availability. In those cases, instead of "WT" they were called "controls."

### Protein isolation and SDS–PAGE

The whole embryo or E12.5 brains frozen at –80°C were washed with ice-cold 1× PBS and homogenized using a Fisher handheld Maximizer homogenizer (Thermo Fisher Scientific) in ice-cold lysis buffer (1 mM ethylene glycol tetraacetic acid [EGTA], 1 mM EDTA, 20 mM Tris base, 1% Triton, 150 mM sodium chloride, 1× cOmplete, Mini, EDTA-free Protease Inhibitor Cocktail [Roche], and 1× PhosSTOP [Roche]). The suspension was centrifuged for 15 min at 120,00$g$ to separate the proteins from the cell debris. The supernatant containing the protein lysate was collected and stored at –80°C. Approximately 8–10 $\mu$g of protein lysate was loaded in each well of a 5–15% SDS–PAGE. The proteins on gels were then transferred overnight on a polyvinylidene fluoride (PVDF) membrane (Bio-Rad) in a buffer containing 25 mM Tris, 190 mM glycine, and 20% methanol. All membranes were blocked at RT for 1 h in Tris-buffered saline (TBST) containing 1 M Tris, 290 mM NaCl, 0.1% Tween-20, pH 7.4, and 5% non-fat dry milk. PVDF membranes were incubated overnight at 4°C with 5% BSA and primary antibodies (listed in Table 1). Membranes were washed five times for 5 min each in TBST before adding the appropriate horseradish peroxidase–labeled secondary antibody (Jackson ImmunoResearch) in a 1:10,000 dilution in TBST with 5% non-fat dry milk. Membranes were shaken slowly at RT for 1 h while incubating with secondary antibody, followed by five washes for 5 min each with TBST. Membranes were treated with a Bio-Rad Western blotting kit (Bio-Rad) and developed using ChemiDoc machines (Bio-Rad). Bands were quantified by densitometry using ImageLab software v.6.0.0 (Bio-Rad). For Stain-Free gels (Bio-Rad), low-fluorescence PVDF membranes (Bio-Rad) were used.

### Primary neuroepithelial cell culture

For the primary neuroepithelial cell culture, embryos at E9.5-10.5 were harvested from plug-checked dams and rinsed in ice-cold 1x PBS before hindbrain–spinal neural tube was isolated from the embryos by shearing the neighboring tissues away with forceps (64). The remaining embryonic tissue was used for PCR genotyping. Isolated neural tube tissues were placed in 300 $\mu$l 1x Trypsin-LE (Thermo Fisher Scientific) for 5 min at 37°C and pipetted up and down to dissociate neuroepithelial cells. This procedure was repeated up to three times for complete breakdown of the neural tube. Cells in Trypsin-LE were inactivated with an equal volume of serum-free culture medium and centrifuged at 0.3$g$ for 5 min to pellet cells. Cells were resuspended with serum-free medium and placed on prepared 1.5-mm coverslips for 24–48 h before proceeding to immunofluorescent imaging. The serum-free culture medium was prepared as detailed previously (64).

### Creation of short-hairpin RNA lentivirus for depletion of SLMAP3

To stably deplete SLMAP3 expression in NE-4-C cells, we used lentivirus vectors previously described (39). Briefly, we designed two shRNAs (S4 and S5) targeting sequences from exon 22 (Ensembl reference transcript ENSMUST00000139075.8) of SLMAP in mouse, and a control scramble (SC). These sequences are as follows: S4-5'GCAATCAATCACAGATGAGCTCAAA, S5-5'GCTGCTGCGAGAGAAAGGAAA-TAAT, and SC-5'AGGATAAGCGTCAACGAATAGGTGA. Lentiviruses were packaged with S4, S5, and scramble control (Sc) sequences, as previously described (65). Reverse transduction was performed in NE-4C cells following the Addgene protocol by using 25,000 cells per cm$^2$ of growth area and considering an MOI of 200 (66). Selection for successfully transduced cells was carried out by treatment of 3 $\mu$g/ml puromycin for 4 d.

### Maintenance, differentiation, TUNEL, and microtubule nucleation assays in NE-4C cells

NE-4C cells (#CRL-2925; ATCC) were grown at 37°C and 5% $CO_2$ in 1x Eagle's Minimum Essential Medium (Wisent) supplemented with 10% FBS (Wisent), 1x antibiotic–antimycotic (Thermo Fisher Scientific), and 2 mM L-glutamine (Invitrogen). The NE-4C cell line examined was negative for mycoplasma contamination. To induce differentiation, transduced NE-4C cells were incubated with 10 $\mu$M retinoic acid (MilliporeSigma) for >96 h (67, 68). Retinoic acid storage, handling, and dilutions were followed as per the manufacturer's protocol. For apoptosis analysis, we used TUNEL Assay Kit from Biotium (cat# 30064) and followed the manufacturer's protocol. The microtubule nucleation assay was performed as described previously (40), where cells were treated with 8.3 $\mu$M of nocodazole for 2 h at 37°C. After this time, cells were washed five times with cold DMEM, and then incubated at 37°C for 45 s with saponin buffer, to clear free tubulin, followed by 45 s with normal medium to allow microtubule growth.

### Statistical analyses

The statistical analysis was performed using GraphPad Prism software version 5 for Windows (GraphPad Software). All comparisons between WT or KO groups were analyzed using a two-tailed $t$ test. To analyze three groups or more, we used one-way ANOVA with Tukey's post hoc test. All error bars presented in graphs are represented using the standard error of the mean.

## Data Availability

RNA-seq data were previously obtained (12) and deposited in NCBI's Gene Expression Omnibus with the GEO Series accession number: GSE230748 (69). To generate heatmaps, the RNA-sequencing data were uploaded to the Galaxy web platform (70). The data from IP-MS with E12.5 mouse brain lysates were recently reported (12) and deposited to the ProteomeXchange Consortium via the PRIDE partner repository with the dataset identifiers PXD041687.

# Supplementary Information

# Acknowledgements

We are grateful for the assistance of all the facilities involved in this study: StemCore Laboratories Genomics Core Facility (RRID:SCR_012601); Ottawa Bioinformatics Core Facility (RRID:SCR_022466); Cell Biology and Image Acquisition Core (RRID: SCR_021845); Louise Pelletier Histology Core Facility (RRID: SCR_021737); and the Montreal Clinical Research Institute (IRCM, Quebec, Canada). This project was funded with the grant 220996-151999 from the Canadian Institutes of Health Research (CIHR) to BS Tuana.

## Author Contributions

T Rehmani: conceptualization, resources, formal analysis, validation, investigation, visualization, methodology, and writing—original draft.

AP Dias: conceptualization, resources, formal analysis, visualization, methodology, and writing—original draft.

BD Applin: formal analysis, investigation, visualization, and writing—original draft.

M Salih: resources, investigation, and methodology.

BS Tuana: conceptualization, supervision, funding acquisition, project administration, and writing—review and editing.

## Conflict of Interest Statement

The authors declare that they have no conflict of interest.

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
