## [Reviewer comments · Life Science Alliance]

Life Science Alliance

SLMAP3 is essential for neurulation through mechanisms involving cytoskeletal elements, ABP and PCP

Taha Rehmani, Ana Dias, Billi Applin, Maysoon Salih, and Balwant Tuana

DOI: <https://doi.org/10.26508/lsa.202302545>

Corresponding author(s): Balwant Tuana, University of Ottawa

Review Timeline:

Submission Date:	2023-12-19
Editorial Decision:	2024-01-19
Revision Received:	2024-08-29
Editorial Decision:	2024-09-20
Revision Received:	2024-09-23
Editorial Decision:	2024-09-24
Revision Received:	2024-09-25
Accepted:	2024-09-26

Transaction Report:

January 19, 2024

Re: Life Science Alliance manuscript #LSA-2023-02545-T

Prof. Balwant S Tuana
University of Ottawa
Cellular and Molecular Medicine
451 Smyth Rd
Ottawa, ONT K1H8M5
Canada

Dear Dr. Tuana,

Thank you for submitting your manuscript entitled "SLMAP3 is essential for neurulation through mechanisms that impact centrosomal and Golgi dynamics" to Life Science Alliance. The manuscript was assessed by expert reviewers, whose comments are appended to this letter. We invite you to submit a revised manuscript addressing the Reviewer comments.

Thank you for this interesting contribution to Life Science Alliance. We are looking forward to receiving your revised manuscript.

Sincerely,

B. MANUSCRIPT ORGANIZATION AND FORMATTING:

Reviewer #1 (Comments to the Authors (Required)):

This paper describes the knockout of SLMAP3 which produces lethality in mice around birth, with some cases of neural tube defects (NTDs), including the severe anomaly craniorachischisis (CR). Molecular analysis shows an effect on planar cell polarity and apico-basal polarity in SLMAP3 KOs, and this can be traced to a defect of centrosomal and Golgi dynamics.

This paper is interesting in the sense that it is a description of a new genetic cause of NTDs in mice, alongside the > 300 others that have been described. The novelty of this study's contribution to understanding NTDs in either animals or humans is unclear, however. Moreover, there are many errors and inadequacies of the developmental biology aspects of this study, which require action by the authors.

1. One feels that the authors are over-interpreting the significance of their findings. For example, they say (lines 342-344): "RNA-Seq analysis indicated that target genes commonly associated with neurulation and craniorachischisis were not significantly altered by SLMAP3 loss, implying that NTD phenotype was not occurring via classical mechanisms and suggest a unique role for SLMAP3 during neurulation". What are these "classical mechanisms" that do not apply to SLMAP3 mutants? Faulty PCP signaling is the best known cause of CR, disrupted apico-basal polarity is well known to cause the defect (e.g. via *Scrb*), and Golgi trafficking is strongly implicated (e.g. via *Sec24b*). Disordered actin turnover is a well known cause of NTDs (e.g. via *Cofilin* knockout, Gurniak, 2005, *Dev Biol.* 278:231-41). This seems to cover all the abnormalities that the authors find in SLMAP3 KOs and so I fail to see how the authors have demonstrated a "unique" mechanism.
2. There is also the issue of the penetrance of NTDs, especially CR. On lines 324-326 the authors say that all KO embryos at E8.5 and subsequent stages were significantly shorter and wider than WT controls (suggesting a convergent extension, CE, defect). Yet only a proportion developed CR. On lines 311-312, it says that 1 out of 3 of SLMAP3-null embryos had this defect, whereas Fig 2C seems to show 9/18 with the defect, which is 1 out of 2. Which is correct? This is in contrast to PCP genes like *Vangl2* that produce 100% CR (in *Lp* mutant) and *Sec24b* knockouts that have 100% CR. What is the authors' explanation for this dramatic difference in fate of individual embryos? Can they look at the severity of alterations in their molecular markers in individual embryos that have different developmental fates?
3. Lines 306-307. "At postnatal day (P)0, SLMAP3-null pups were stillborn, displaying signs of late embryonic lethality". The embryonic period ends at E13.5, after which the fetal period occurs to birth. If stillborn (i.e. dead at P0), this is late fetal or neonatal lethality, not embryonic.
4. Lines 399-400. "During neurulation, the neural plate can express varying cadherins at different developmental stages [49]. At E7.5 it expresses E-cadherin (E-cad)". This is incorrect. At E7.5, the epiblast expresses *Ecad* but it has not yet differentiated towards neuroepithelium. Once neural plate is formed, it is *Ecad* negative. It is not clear where *Ecad* is being measured in Fig 5C, to achieve the differences that are graphed below. At E9.5, the bright *Ecad* expression in WT is in the surface (non-neural) ectoderm, whereas there is minimal expression in WT or KO neural tube. This point applies to most other expression data - the authors need to indicate on images where expression was measured from.
5. Lines 533-534. "Our data here defines a critical role of SLMAP3 in embryonic development, specifically in developing the neural tube since its loss results in failure to close leading to craniorachischisis". As mentioned above, this is inaccurate, as only a proportion of embryos show this defect.
6. Lines 553-554. "SLMAP3 may be a novel biomarker for NTDs". What does this mean. Are the authors suggesting that SLMAP3 mutations lead to human NTDs? Have they examined patient DNA for coding region mutations in the gene? Or perhaps they are thinking of another meaning of 'biomarker' in this context?
7. Fig 2B. KO embryos are said to have "facial retrusion", but this is not explained morphologically, and the graph simply says "craniofacial". More explanation, preferably with dimensions of the facial features indicated by arrows on images, is needed. The "truncated tail" is also not clearly visible in the figure. PCP and *Sec24b* mutants with CR typically have curled tails - is that present in the KO embryos/fetuses?

8. Fig 2C. KO embryos have n = 18, but phenotypes of only 11 embryos is listed. Were the other 7 embryos normal? How was this judged? What stage have these 'embryos' reached? Are they P0 as in Fig 2B? Phenotypes of all embryos/fetuses/newborns needs to be shown in the Table. Also, the way it is presented, KO seems to be an alternative to the 3 phenotypes. Please re-design the table.

9. Fig 3A. "Representative wholemount images of Wt and KO embryos during prenatal development (E8.5-E13.5)". I do not see how the KO embryos as shown can be 'representative' when Fig 2C says that 7 of 18 do not have a phenotype (apart from a short tail?). The spectrum of phenotypes of KO embryos needs to be presented at each stage studied.

10. Fig 3D. What are the line/angle diagrams below the bar graph? The explanation in the legend seems to be "... changes in E9.5 neural groove aperture" but this is not clear. Are the angles averaged over different sections/embryos in which case a standard deviation would apply?

11. Fig 5 legend. This is very repetitive and there is no need to repeat "Representative" for each part of the figure.

12. In general, text on many figures (e.g. graphs in Fig 2A, 9D) is so small as to be impossible to read. Moreover, the text has many typos, inconsistent use of nomenclature and grammatical errors throughout the manuscript that require attention.

Reviewer #2 (Comments to the Authors (Required)):

This manuscript delves into the role of SLMAP3 during neurulation, a critical process in vertebrate embryo development. Given that abnormal regulation of convergent and extension movement can result in defects such as neural tube closure and short axis elongation, understanding the molecular mechanisms underlying neural tube closure becomes crucial. This study demonstrated that the knockout of SLMAP3 in mice results in an open neural tube and craniorachischisis. Molecular analysis suggests that SLMAP3 is implicated in the regulation of Wnt-PCP signaling, centrosome dynamics, and Golgi dynamics. However, there are concerns regarding the conclusions drawn, as some seem to be based on presumptive leaps rather than direct experimental evidence presented in the manuscript. Additionally, there are perceived deficiencies in describing the methods employed for certain aspects of the analysis.

The following concerns have been identified:

1. In Figures 2E and F, the author examined Hippo signaling using whole embryo lysates from SLMAP3 knockout embryos. It is noteworthy that SLMAP3 knockout mice displayed specific defects in the neural plate development. Given that the neural plate constitutes a relatively small portion of the whole embryo, both Western blot and RNA sequencing analyses may not accurately represent Hippo signaling as it relates to the neural tube closure defect phenotype. Consequently, it is recommended to conduct the Hippo signaling analysis in the neural plate specifically or NEP (primary neuroepithelial cells).

2. Following up with point 1 above, for an accurate analysis of SLMAP3 knockout defects in Figure 3E, it is recommended to perform RNA-seq analysis using the neural plate instead of the whole embryo.

3. In the heatmap of Figure 3E, both Dvl1 and Dvl2 appear to be elevated in SLMAP3 KO embryos, and Dvl3 is not decreased (actually increased in KO3 embryos). These findings are inconsistent with the Western blot results in Figure 4C. The author should provide clarification on why E12.5 brains, instead of neural plate, were chosen for the analysis of PCP signaling and address the discrepancy between the RNA-seq results and Western blot results.

4. In Figure 5C, unlike N-cadherin, the immunostaining results indicate that the E-cadherin level at the apical area of neural plates appears to be similar. However, the histograms suggest a significant reduction in SLMAP3 KO embryos. While there is a robust E-cadherin signal in the E9.5 embryo, it's crucial to note that the outer layer corresponds to the skin, not the neuroepithelial layer. To enhance clarity, please provide a more explicit interpretation and details on the analysis methods employed for this examination.

5. In Figures 8 and 9, the mislocalization of pericentrin, Golgi fragmentation, and the localization of Scribble, AJs, and TJs should be examined in the neural plate of the developing embryo. Have similar analyses been conducted using embryos rather than cultured isolated neural tube cells? Were consistent results observed in the neural plate of the developing embryo? If there were technical challenges, providing a clearer understanding of the study's limitations and identifying questions that remain to be addressed in future work would be valuable.

6. In Figure 10, the data regarding the impact of SLMAP3 on the differentiation of NE-4C cells appears to be out of focus from the main focus of the study. This research primarily addresses how SLMAP3 regulates PCP signaling, leading to an open neural tube and craniorachischisis. The connection between the differentiation defect and the neural tube closure defect phenotype is unclear. To maintain focus on the main theme, it is recommended either to remove Figure 10 or relegate it to the Supplementary section.

Minor points:

1. In Fig2D, the N-terminus of the protein is typically on the left side, while the C-terminus is positioned on the right side.
2. In Fig2F, Fig3E, and Fig4D, to improve the convenience of data interpretation, it is recommended to rearrange the heatmap to present the comparison as WT vs KO.
3. Fig 4C, providing a clearer data interpretation regarding the relationship between Dvl expression levels and the regulation of Rock2, JNK, or cofilin would be beneficial.
4. In Figure 5B, the apical-basal polarity of PKC-zeta and Nestin should be analyzed at the single-cell level in the outer layer of the neural plate. High-magnification images of the outer layer of the neural plate are necessary to substantiate the author's hypothesis.

Reviewer #3 (Comments to the Authors (Required)):

In this manuscript, the authors report a role of SLMAP3 in neural tube closure during development in mice. Loss of SLMAP3 causes neural tube defects and disrupts expression and localization of planar cell polarity (PCP) and apical-basal polarity (ABP) proteins. Through immunoprecipitation and mass spectrometry analysis, they find that SLMAP3 interacts with Scribble and components of centrosome and Golgi. Finally, they show that knockdown of SLMAP3 in NE-4C mouse neuroepithelial cells affects differentiation. Although the phenotypes of knockout mice suggest that SLMAP3 may be involved in neural tube morphogenesis, the conclusions that it regulates cell polarity are not strongly supported by experimental data. Several results presented in the manuscript also lack consistency. In its present form, the manuscript is not suitable for publication.

Major concerns:

1. Loss of SLMAP3 affects cell differentiation. Thus, it is unclear whether the defective morphology of neural epithelial cells associated with disrupted expression and localization of PCP and ABP proteins is a direct consequence due to loss of SLMAP3 or represents an indirect consequence of impaired cell differentiation. The authors examined neural differentiation markers in NE-4C cells. This should be also done in the neuroepithelia from whole embryos.
2. There is a lack of focus in the manuscript to assess the mechanism by which SLMAP3 regulates neural tube closure. The authors superficially examined many aspects of cell polarity, and data were put together without a clear link. They need to provide an in-depth mechanistic analysis of SLMAP3-regulated cellular and molecular processes.
3. As I can see, the authors only examined expression of PCP-related proteins in brain extracts by western blot, but they did not show mis-localization of these proteins in neuroepithelial cells. There is no explanation on the differential regulation of Dvl2 and Dvl3 following loss of SLMAP3. Dvl2 plays a predominant role in convergent extension of the neural tube in different vertebrates. It is thus necessary to examine its localization by immunofluorescence, if the authors want to focalize their study on the regulation of PCP.
4. In contrast to the statement in the abstract and results sections, the authors did not show the interaction of SLMAP3 with Glogb1, Golga3 and Kif5c by coimmunoprecipitation, and their localization by immunofluorescence. In figure 6C, why the band immunoprecipitated by anti-SLMAP3 antibody shows a different size than Nestin in other conditions? The authors should include protein size markers in all western blots.
5. The results in figure 5C and D do not seem to be consistent. In figure 5C, there is a reduced fluorescent intensity of N-cadherin and E-cadherin in the neural tube of SLMAP3-KO mice. However, figure 5D shows increased expression of E-cadherin in brain extracts of SLMAP3-KO mice. In addition, it is unclear whether figure 5D shows two independent samples at the same stage or it shows samples from two different stages.
6. Staining of Pericentrin and GML130 is problematic. In lines 445-447, the authors state that the Golgi apparatus appears elongated in Wt NT but was fragmented in SLMAP3-KO with decrease in size in E8.5 and E9.5. However, it is not possible to see these in figure 7, particularly at stage E9.5. In figure 8C and 10D, GM130 staining seems to be localized in the nuclei, not in the cytoplasm. In figure 7A and in the Sc condition of figure 10D, staining of Pericentrin seems to be present in the cytoplasm. Another problem is that Pericentrin staining is reduced in NEPCs (figure 8B and C) but it is apparently increased in NE-4C cells (figure 10D) following loss of SLMAP3.
7. It seems that Golgi apparatus show fragmentation in NEPCs. However, they appear intact in NE-4C cells. What is the explanation for these differences? The conclusion that Golgi fragmentation impacts NE-4C differentiation is not supported by experimental evidence.
8. Immunofluorescent staining of fixed cells does not allow to determine the dynamics of centrosome and Golgi. The authors should use live imaging to monitor dynamic changes of these organelles.
9. Since there is inconsistent isolation of primary neuroepithelial cells, why the authors use them to analyze cell size, mis-localization of Pericentrin, and Golgi fragmentation? There is a possibility that these cells change polarity during culture. Curiously, the localization of basal marker Nestin appears more restricted in SLMAP3-KO cells, as shown in figure 8B.
10. Lines 409-410, "These data indicate that SLMAP3 regulates ABP within the neuroepithelial tissue for proper distribution of PCP components". There is no data supporting this conclusion. There is no evidence showing the relationship between PCP, ABP, and centrosomal and Golgi components in the context of SLMAP3 KO.
11. The analysis of NE-4C cell differentiation following SLMAP3 knockdown is very superficial. What is the fate of cells with decreased expression of Sox2 and Pax6? Is there reduced proliferation? Do these cells undergo apoptosis?

12. The manuscript is not well structured. There are 11 figures, some figures can be combined, and some data, such as the analysis of Hipo signaling shown in figure 2, can be presented as supplementary figures. The authors need to thoroughly re-organize their manuscript and present consistent results.

Minor concerns:

1. Table 1 can be presented as a supplementary table.
2. In several figures, the labels are barely legible. Font size should be increased.
3. Developmental stages of mouse embryos should be written in a consistent manner. For example, E8.5, not e8.5.
4. Bcatenin should be β -catenin.
5. Line 499, 10 μm should be 10 μM .
6. ZO1 localization should be examined using embryo sections.
7. The manuscript is not well written. There are many typos and grammatical errors.

Reviewer #1 (Comments to the Authors (Required)):

This paper describes the knockout of SLMAP3 which produces lethality in mice around birth, with some cases of neural tube defects (NTDs), including the severe anomaly craniorachischisis (CR). Molecular analysis shows an effect on planar cell polarity and apico-basal polarity in SLMAP3 KOs, and this can be traced to a defect of centrosomal and Golgi dynamics.

This paper is interesting in the sense that it is a description of a new genetic cause of NTDs in mice, alongside the > 300 others that have been described. The novelty of this study's contribution to understanding NTDs in either animals or humans is unclear, however. Moreover, there are many errors and inadequacies of the developmental biology aspects of this study, which require action by the authors.

Thanks for taking the time to review this manuscript and the excellent feedback! We have tried to address all the issues raised, and which have made the manuscript clearer and focused.

1. One feels that the authors are over-interpreting the significance of their findings. For example, they say (lines 342-344): "RNA-Seq analysis indicated that target genes commonly associated with neurulation and craniorachischisis were not significantly altered by SLMAP3 loss, implying that NTD phenotype was not occurring via classical mechanisms and suggest a unique role for SLMAP3 during neurulation". What are these "classical mechanisms" that do not apply to SLMAP3 mutants? Faulty PCP signaling is the best known cause of CR, disrupted apico-basal polarity is well known to cause the defect (e.g. via *Scrb*), and Golgi trafficking is strongly implicated (e.g. via *Sec24b*). Disordered actin turnover is a well known cause of NTDs (e.g. via *Cofilin* knockout, Gurniak, 2005, *Dev Biol.* 278:231-41). This seems to cover all the abnormalities that the authors find in SLMAP3 KOs and so I fail to see how the authors have demonstrated a "unique" mechanism.

We agree and removed the word "unique".

2. There is also the issue of the penetrance of NTDs, especially CR. On lines 324-326 the authors say that all KO embryos at E8.5 and subsequent stages were significantly shorter and wider than WT controls (suggesting a convergent extension, CE, defect). Yet only a proportion developed CR. On lines 311-312, it says that 1 out of 3 of SLMAP3-null embryos had this defect, whereas Fig 2C seems to show 9/18 with the defect, which is 1 out of 2. Which is correct? This is in contrast to PCP genes like *Vangl2* that produce 100% CR (in *Lp* mutant) and *Sec24b* knockouts that have 100% CR. What is the authors' explanation for this dramatic difference in fate of individual embryos? Can they look at the severity of alterations in their molecular markers in individual embryos that have different developmental fates?

The current Figure 1C is identifying the penetrance of the phenotypes that occur due to a loss of SLMAP3. Most SLMAP3-KO embryos (18) do not exhibit NTD, but all display craniofacial retrusion, growth retardation and short tail. Further, another 9 of the SLMAP3-KO embryos exhibited CR, 1 embryo exhibited spina bifida and 1 exencephaly. All together, the total number of SLMAP3-KO embryos is 29 and 9/29 (~31%) with CR. We have clarified this in the legend of Figure 1C. Although there could be genetic modifiers involved that may explain the differences observed in the phenotypes, we have not mapped them yet for the case of SLMAP, which would be interesting for future studies.

3. Lines 306-307. "At postnatal day (P)0, SLMAP3-null pups were stillborn, displaying signs of late embryonic lethality". The embryonic period ends at E13.5, after which the fetal period occurs to birth. If stillborn (i.e. dead at P0), this is late fetal or neonatal lethality, not embryonic.

Thanks for the suggestion. We corrected it in the text.

4. Lines 399-400. "During neurulation, the neural plate can express varying cadherins at different developmental stages [49]. At E7.5 it expresses E-cadherin (E-cad)". This is incorrect. At E7.5, the epiblast expresses Ecad but it has not yet differentiated towards neuroepithelium. Once neural plate is formed, it is Ecad negative. It is not clear where Ecad is being measured in Fig 5C, to achieve the differences that are graphed below. At E9.5, the bright Ecad expression in WT is in the surface (non-neural) ectoderm, whereas there is minimal expression in WT or KO neural tube. This point applies to most other expression data - the authors need to indicate on images where expression was measured from.

Thanks for bringing this up and you are correct. The expression data indicates only N-Cad in NT, which is not altered in terms of its distribution in SLMAP3-KO animals (Figure S3A).

5. Lines 533-534. "Our data here defines a critical role of SLMAP3 in embryonic development, specifically in developing the neural tube since its loss results in failure to close leading to craniorachischisis". As mentioned above, this is inaccurate, as only a proportion of embryos show this defect.

Thanks for the suggestion. We have replaced this sentence with: "Our data here defines a critical role of SLMAP3 in neural tube, since its loss causes some embryos to exhibit NTDs, including incomplete closure and craniorachischisis."

6. Lines 553-554. "SLMAP3 may be a novel biomarker for NTDs". What does this mean. Are the authors suggesting that SLMAP3 mutations lead to human NTDs? Have they examined patient DNA for coding region mutations in the gene? Or perhaps they are thinking of another meaning of 'biomarker' in this context?

We meant that SLMAP should be examined in human NTDs as could provide a new link in affected patients. This statement has been replaced with "Since SLMAP3 loss leads to these severe NT deficits, it may be worthwhile interrogating human samples of NTDs for any links to SLMAP".

7. Fig 2B. KO embryos are said to have "facial retrusion", but this is not explained morphologically, and the graph simply says "craniofacial". More explanation, preferably with dimensions of the facial features indicated by arrows on images, is needed. The "truncated tail" is also not clearly visible in the figure. PCP and Sec24b mutants with CR typically have curled tails - is that present in the KO embryos/fetuses?

We have included in the text and Figure legend (now Figure 1B) stunted snout to better describe the phenotype. We have also indicated with arrow in Figure 1B the truncated tail present in SLMAP3^{-/-} animals.

8. Fig 2C. KO embryos have n = 18, but phenotypes of only 11 embryos is listed. Were the other 7 embryos normal? How was this judged? What stage have these 'embryos' reached? Are they P0 as in Fig 2B? Phenotypes of all embryos/fetuses/newborns needs to be shown in the Table. Also, the way it is presented, KO seems to be an alternative to the 3 phenotypes. Please re-design the table.

Sorry for the confusion. As mentioned earlier, the number of KO is n=29. All embryos exhibited embryonic lethality. The table has been relabelled for clarity.

9. Fig 3A. "Representative wholemount images of Wt and KO embryos during prenatal development (E8.5-E13.5)". I do not see how the KO embryos as shown can be 'representative' when Fig 2C says that 7 of 18

do not have a phenotype (apart from a short tail?). The spectrum of phenotypes of KO embryos needs to be presented at each stage studied.

Sorry for the misunderstanding. From E8.5-E10.5, it is hard to determine which KO embryos will undergo CR or not. All the KO embryos are smaller in size and exhibit lethality.

10. Fig 3D. What are the line/angle diagrams below the bar graph? The explanation in the legend seems to be "... changes in E9.5 neural groove aperture" but this is not clear. Are the angles averaged over different sections/embryos in which case a standard deviation would apply?

The previous Figure 3D, now Figure 1G has the diagrams that represent the average of neural aperture groove. We included an additional graphic with the measure of the angles in SLMA3^{-/-} and WT embryos for clarity.

11. Fig 5 legend. This is very repetitive and there is no need to repeat "Representative" for each part of the figure.

Thank you, we have modified this in the legend of all figures.

12. In general, text on many figures (e.g. graphs in Fig 2A, 9D) is so small as to be impossible to read. Moreover, the text has many typos, inconsistent use of nomenclature and grammatical errors throughout the manuscript that require attention.

We have corrected the errors in the entire manuscripts and figures.

Reviewer #2 (Comments to the Authors (Required)):

This manuscript delves into the role of SLMAP3 during neurulation, a critical process in vertebrate embryo development. Given that abnormal regulation of convergent and extension movement can result in defects such as neural tube closure and short axis elongation, understanding the molecular mechanisms underlying neural tube closure becomes crucial. This study demonstrated that the knockout of SLMAP3 in mice results in an open neural tube and craniorachischisis. Molecular analysis suggests that SLMAP3 is implicated in the regulation of Wnt-PCP signaling, centrosome dynamics, and Golgi dynamics. However, there are concerns regarding the conclusions drawn, as some seem to be based on presumptive leaps rather than direct experimental evidence presented in the manuscript. Additionally, there are perceived deficiencies in describing the methods employed for certain aspects of the analysis. **Thanks for the detailed review and we have tried to address the concerns indicated.**

The following concerns have been identified:

1. In Figures 2E and F, the author examined Hippo signaling using whole embryo lysates from SLMAP3 knockout embryos. It is noteworthy that SLMAP3 knockout mice displayed specific defects in the neural plate development. Given that the neural plate constitutes a relatively small portion of the whole embryo, both Western blot and RNA sequencing analyses may not accurately represent Hippo signaling as it relates to the neural tube closure defect phenotype. Consequently, it is recommended to conduct the Hippo signaling analysis in the neural plate specifically or NEP (primary neuroepithelial cells).

SLMAP3 is the only isoform expressed in embryonic brain and its loss was found to have no impact on Hippo signaling. We have added this as Figure 2D. Further, we added in supplemental Figure S2A an immunofluorescent image of YAP staining and saw no discernable differences in localization or expression in SLMAP3-deficient E9.5 neural tube when compared to wildtype. Finally, due to reduced body length of SLMAP3 KO embryos, RNA-seq and Western Blot with whole embryos samples would have indicated Hippo abnormalities if that was the case.

2. Following up with point 1 above, for an accurate analysis of SLMAP3 knockout defects in Figure 3E, it is recommended to perform RNA-seq analysis using the neural plate instead of the whole embryo.

Although that would be very interesting, isolating the neural plate was quite difficult and did not provide enough tissue for downstream analysis. Besides, as the whole embryo is underdeveloped and defective (Figure 1B and 1D), analysis with whole embryo lysates would still shows any aberrant gene expression.

3. In the heatmap of Figure 3E, both Dvl1 and Dvl2 appear to be elevated in SLMAP3 KO embryos, and Dvl3 is not decreased (actually increased in KO3 embryos). These findings are inconsistent with the Western blot results in Figure 4C. The author should provide clarification on why E12.5 brains, instead of neural plate, were chosen for the analysis of PCP signaling and address the discrepancy between the RNA-seq results and Western blot results.

Regarding the DVLS in the heatmap, sorry for the confusion. We have modified the heatmaps to display the results more clearly. About the western blot analysis, because it is difficult to obtain enough sample from neural plates for such analysis we used brains instead.

4. In Figure 5C, unlike N-cadherin, the immunostaining results indicate that the E-cadherin level at the apical area of neural plates appears to be similar. However, the histograms suggest a significant reduction

in SLMAP3 KO embryos. While there is a robust E-cadherin signal in the E9.5 embryo, it's crucial to note that the outer layer corresponds to the skin, not the neuroepithelial layer. To enhance clarity, please provide a more explicit interpretation and details on the analysis methods employed for this examination. Thanks for bringing this up. We have removed the E-cadherin staining and quantification from the neural plate, as this tissue is only expected to express N-cadherin at the developmental stages that we analyzed. We have reanalyzed N-cadherin staining across the neural tube, but no changes were detected (Figure S3A). We used plot profile analysis from Fiji by drawing lines from basal to the apical region of the neural tube and quantifying the staining intensities. We have now included imaging processing procedures for all images in the manuscript in the materials and methods section, Histological and Immunofluorescent analysis.

5. In Figures 8 and 9, the mislocalization of pericentrin, Golgi fragmentation, and the localization of Scribble, AJs, and TJs should be examined in the neural plate of the developing embryo. Have similar analyses been conducted using embryos rather than cultured isolated neural tube cells? Were consistent results observed in the neural plate of the developing embryo? If there were technical challenges, providing a clearer understanding of the study's limitations and identifying questions that remain to be addressed in future work would be valuable.

Thank you for the suggestion and we have now included the data in NT with significant changes in F-actin (Figure 4B and C), aPKC ζ (Figure 4D), ZO-1 (Figures 4E and 4F), Nestin (Figure 4G) and γ -tubulin (Figure 5D). Further, reanalysis of the data in NE-4C cells indicated no changes in cytoskeleton components (Figures 7A, 7B, Figure S5A and S5B), pericentrin or Golgi (Figures S6A and S6B), indicating that SLMAP3 impacts these components *in vivo* only. One of the major challenges of studying PCP in culture is that external cues and three dimensional architecture, which have major impact on this process cannot be properly reproduced. Given that SLMAP3 loss *in vivo* impacted PCP, further supports the specificity of the system.

6. In Figure 10, the data regarding the impact of SLMAP3 on the differentiation of NE-4C cells appears to be out of focus from the main focus of the study. This research primarily addresses how SLMAP3 regulates PCP signaling, leading to an open neural tube and craniorachischisis. The connection between the differentiation defect and the neural tube closure defect phenotype is unclear. To maintain focus on the main theme, it is recommended either to remove Figure 10 or relegate it to the Supplementary section.

Thank you for the suggestion we have removed this data from the manuscript.

Minor points:

1. In Fig2D, the N-terminus of the protein is typically on the left side, while the C-terminus is positioned on the right side.

Thanks, we have updated the image, which now is Figure 2A.

2. In Fig2F, Fig3E, and Fig4D, to improve the convenience of data interpretation, it is recommended to rearrange the heatmap to present the comparison as WT vs KO.

Thanks for the suggesting, but this is difficult as we wanted to display a Euclidean clustering to display the similarities in gene expression vs genotype. Further, because the KO samples did not cluster together, it already shows that changes in the gene expression of the process associated with Hippo (Figure S2C) and Craniorachichisis (Figure 1H) were not biologically significant.

3. Fig 4C, providing a clearer data interpretation regarding the relationship between Dvl expression levels and the regulation of Rock2, JNK, or cofilin would be beneficial.

Thanks for the suggestion, but the relationship between Dvls and the downstream proteins ROCK2, JNK and cofilin are already discussed in the result section 2.3. The analysis of the β -catenin is now included in Figure 3F and Figure S3B and S3C to show if the canonical Wnt pathway was affected, which is not the case.

4. In Figure 5B, the apical-basal polarity of PKC-zeta and Nestin should be analyzed at the single-cell level in the outer layer of the neural plate. High-magnification images of the outer layer of the neural plate are necessary to substantiate the author's hypothesis.

Thanks for the suggestion, but we already used lens with 63x magnification, which is standard to investigate localization of proteins within specific compartments. Further, imaging single cells in a tissue is very challenging considering that cells overlap on each other, making it very difficult to visualize their limits. About PKC-zeta and Nestin staining, because cells in the neural tube might be undergoing interkinetic nuclear migration, visualizing a well defined layer of cells is very challenging. To improve our PKC-zeta and Nestin analysis, we remeasured the staining across the neural tube, from basal to apical region, and we have plotted the staining distribution, which now we included as Figure 4D and 4G, respectively.

Reviewer #3 (Comments to the Authors (Required)):

In this manuscript, the authors report a role of SLMAP3 in neural tube closure during development in mice. Loss of SLMAP3 causes neural tube defects and disrupts expression and localization of planar cell polarity (PCP) and apical-basal polarity (ABP) proteins. Through immunoprecipitation and mass spectrometry analysis, they find that SLMAP3 interacts with Scribble and components of centrosome and Golgi. Finally, they show that knockdown of SLMAP3 in NE-4C mouse neuroepithelial cells affects differentiation. Although the phenotypes of knockout mice suggest that SLMAP3 may be involved in neural tube morphogenesis, the conclusions that it regulates cell polarity are not strongly supported by experimental data. Several results presented in the manuscript also lack consistency. In its present form, the manuscript is not suitable for publication. **Thanks for the time to review this manuscript and detailed critique with a view to improve the manuscript. We have attempted to correct the issues raised, and reanalyzed the data which we believe has helped in clarity and improved the text.**

Major concerns:

1. Loss of SLMAP3 affects cell differentiation. Thus, it is unclear whether the defective morphology of neural epithelial cells associated with disrupted expression and localization of PCP and ABP proteins is a direct consequence due to loss of SLMAP3 or represents an indirect consequence of impaired cell differentiation. The authors examined neural differentiation markers in NE-4C cells. This should be also done in the neuroepithelia from whole embryos.

We analyzed Nestin marker, which was expressed in both KO and WT neural tubes.

2. There is a lack of focus in the manuscript to assess the mechanism by which SLMAP3 regulates neural tube closure. The authors superficially examined many aspects of cell polarity, and data were put together without a clear link. They need to provide an in-depth mechanistic analysis of SLMAP3-regulated cellular and molecular processes. **Thank you -great point! We have reorganized the manuscript with a clear thread on how the PCP/APB components were affected in SLMAP3 deficiency, together with the new SLMAP3 interactors including the cytoskeleton components and the changes in their activity/distribution that imply an important role for SLMAP3 in cell polarity. We have provided a new model (Figure 7) that links SLMAP3 with the key components for PCP/APB based on the new data, which supports a signaling role that remains to be fully elucidated**

3. As I can see, the authors only examined expression of PCP-related proteins in brain extracts by western blot, but they did not show mis-localization of these proteins in neuroepithelial cells. There is no explanation on the differential regulation of Dvl2 and Dvl3 following loss of SLMAP3. Dvl2 plays a predominant role in convergent extension of the neural tube in different vertebrates. It is thus necessary to examine its localization by immunofluorescence, if the authors want to focalize their study on the regulation of PCP.

Thanks for your suggestion. We have now included new data on immunofluorescence analysis of Fzd6 and pDVL2, which shows that the colocalization of these key players is dependent on SLMAP3. The data in NTs was not recapitulated in cell culture, implying that PCP/APB is critically dependent on *in vivo* cues, which are absent in cultured conditions.

4. In contrast to the statement in the abstract and results sections, the authors did not show the

interaction of SLMAP3 with Glogb1, Golga3 and Kif5c by coimmunoprecipitation, and their localization by immunofluorescence. In figure 6C, why the band immunoprecipitated by anti-SLMAP3 antibody shows a different size than Nestin in other conditions? The authors should include protein size markers in all western blots.

We agree with this point. We have removed the association of SLMAP with Golgi proteins from the abstract. About Figure 6C, now Figure 5C, the size of nestin band in Anti-SLMAP lane is similar to the other lanes. The additional band with higher molecular weight may correspond to the protein with post-translational modification, such as phosphorylation. Also, we provided protein size markers in our blots as requested.

5. The results in figure 5C and D do not seem to be consistent. In figure 5C, there is a reduced fluorescent intensity of N-cadherin and E-cadherin in the neural tube of SLMAP3-KO mice. However, figure 5D shows increased expression of E-cadherin in brain extracts of SLMAP3-KO mice. In addition, it is unclear whether figure 5D shows two independent samples at the same stage or it shows samples from two different stages.

Sorry about the confusion. In the Figure 5C, now replaced with Figure S3A, N-cadherin expression was equally well expressed in SLMAP3 KO neural tube at E8.5. The E-cadherin staining was removed since it is not expressed in the neural tube at these developmental stages. E-cadherin expression in Figure 5D, now Figure 4H, represents analysis in brain lysates (N=4) and at a different developmental stage (E12.5) and we believe to be associated with delayed differentiation of the brain. Both lanes displayed for each genotype are two biological replicates out of four that we analyzed of E12.5 brain lysates.

6. Staining of Pericentrin and GML130 is problematic.

- (about Figure 7A) In lines 445-447, the authors state that the Golgi apparatus appears elongated in Wt NT but was fragmented in SLMAP3-KO with decrease in size in E8.5 and E9.5. However, it is not possible to see these in figure 7, particularly at stage E9.5.

- In figure 8C and 10D, GM130 staining seems to be localized in the nuclei, not in the cytoplasm.

The reviewer is correct and upon re-examination of the raw data, we found no changes as indicated in new Figures S4A and S4B.

- In figure 7A and in the Sc condition of figure 10D, staining of Pericentrin seems to be present in the cytoplasm. Another problem is that Pericentrin staining is reduced in NEPCs (figure 8B and C) but it is apparently increased in NE-4C cells (figure 10D) following loss of SLMAP3.

As in point above.

7. It seems that Golgi apparatus show fragmentation in NEPCs. However, they appear intact in NE-4C cells. What is the explanation for these differences? The conclusion that Golgi fragmentation impacts NE-4C differentiation is not supported by experimental evidence. This is a good point. We reanalyzed our raw data and no differences in Golgi were found in Neural tube (Figure S4A), NEPs (Figure S5A) and NE-4C cells (Figure S6A and S6B).

8. Immunofluorescent staining of fixed cells does not allow to determine the dynamics of centrosome and

Golgi. The authors should use live imaging to monitor dynamic changes of these organelles. **We agree, we should not have used the word “dynamics”. Still, our reanalysis indicates not changes in Golgi.**

9. Since there is inconsistent isolation of primary neuroepithelial cells, why the authors use them to analyze cell size, mis-localization of Pericentrin, and Golgi fragmentation? There is a possibility that these cells change polarity during culture. Curiously, the localization of basal marker Nestin appears more restricted in SLMAP3-KO cells, as shown in figure 8B. **Although cells could change cell polarity in culture, we would still be able to measure cell size. About Nestin, it is indeed more concentrated in the basal region of SLMAP3^{-/-} neural tubes according to our quantification (Figure 4G).**

10. Lines 409-410, "These data indicate that SLMAP3 regulates ABP within the neuroepithelial tissue for proper distribution of PCP components". There is no data supporting this conclusion. There is no evidence showing the relationship between PCP, ABP, and centrosomal and Golgi components in the context of SLMAP3 KO.

We agree, we overstated this and now tempered the discussion. We did observe that knockout of SLMAP3 affected PCP and ABP components, which is likely associated with the NTDs observed in SLMAP3 embryos.

11. The analysis of NE-4C cell differentiation following SLMAP3 knockdown is very superficial. What is the fate of cells with decreased expression of Sox2 and Pax6? Is there reduced proliferation? Do these cells undergo apoptosis?

We agree with this and removed the data. Still, we conducted a TUNEL assay to quantify cell death in NE-4C cells (Figure 6E now included) and pH3 and Ki67 staining for proliferation (Figure 6C and 6D now included). Although no differences were observed in apoptosis, we did observe a significant decrease in proliferation of NE-4C cells upon induction of differentiation.

12. The manuscript is not well structured. There are 11 figures, some figures can be combined, and some data, such as the analysis of Hippo signaling shown in figure 2, can be presented as supplementary figures. The authors need to thoroughly re-organize their manuscript and present consistent results.

We have reorganized the entire manuscript now, having 8 main figures and 6 supplementary ones. Hippo signaling data needs to remain as a central figure since SLMAP3 has been linked to Hippo signaling in a multitude of studies, although we moved the heatmap to supplementary material. Investigating if Hippo signaling was impacting the NTDs observed in SLMAP3 KO embryos remains important and our data negates a role for SLMAP3 in this process *in vivo*.

Minor concerns:

1. Table 1 can be presented as a supplementary table.

Thank you for the suggestion. Because we already have the heatmaps of Hippo and NTDs related genes, we removed this table.

2. In several figures, the labels are barely legible. Font size should be increased.

We have fixed all images.

3. Developmental stages of mouse embryos should be written in a consistent manner. For example, E8.5, not e8.5.

Thank you for pointing this out, we have made sure to keep the language consistent.

4. Bcatenin should be β -catenin. We have fixed this.

5. Line 499, 10 μm should be 10 μM .
We have corrected it.

6. ZO1 localization should be examined using embryo sections.
We have performed this experiment and added this to the manuscript in Figure 4E and 4F.

7. The manuscript is not well written. There are many typos and grammatical errors.
Thank you for pointing this out, we have gone through the manuscript and have fixed the typos and grammar.

September 20, 2024

Re: Life Science Alliance manuscript #LSA-2023-02545-TR

Prof. Balwant S Tuana
University of Ottawa
Cellular and Molecular Medicine
451 Smyth Rd
Ottawa, ONT K1H8M5
Canada

Dear Dr. Tuana,

Thank you for submitting your revised manuscript entitled "SLMAP3 is essential for neurulation through mechanisms involving cytoskeletal elements, ABP and PCP" to Life Science Alliance. The manuscript has been seen by the original reviewers whose comments are appended below. While the reviewers continue to be overall positive about the work in terms of its suitability for Life Science Alliance, some important issues remain.

Our general policy is that papers are considered through only one revision cycle; however, given that the suggested changes are relatively minor, we are open to one additional short round of revision. Please note that I will expect to make a final decision without additional reviewer input upon re-submission.

Please submit the final revision within one month, along with a letter that includes a point by point response to the remaining reviewer comments.

To upload the revised version of your manuscript, please log in to your account: <https://lsa.msubmit.net/cgi-bin/main.plex>
You will be guided to complete the submission of your revised manuscript and to fill in all necessary information.

B. MANUSCRIPT ORGANIZATION AND FORMATTING:

Sincerely,

Reviewer #2 (Comments to the Authors (Required)):

I thank the authors for their thorough and comprehensive responses to my previous review; all of my concerns/comments have been addressed.

Reviewer #3 (Comments to the Authors (Required)):

In this revised manuscript, the authors have addressed some of the concerns raised in my previous review. However, there are still a number of issues that need to be clarified by thorough discussion and additional analyses.

1. The authors state that they used Nestin to monitor neuroepithelial cell differentiation. Is Nestin considered as a marker of differentiation? The expression of Nestin was only examined at E9.5 (Figure 4G), which seems to show a decrease or absence at the apical side. The expression of N-cadherin is associated with neuroepithelial cell differentiation, but this was also examined at a single stage (E8.5 in Figure S3A), and there are no changes in its distribution in the neural tube at this stage. Therefore, it is unclear whether the loss of SLMAP3 affects neuroepithelial cell differentiation.
2. The authors suggest that "SLMAP3 may have a mode of action in neural tube development through post-transcriptional mechanisms". How does SLMAP3 post-transcriptionally regulate neural tube development? This should be discussed by presenting arguments supporting the hypothesis.
3. In Figure 1C and D, I suppose that "KD" refers to heterozygous mutants. The use of "KD" is not appropriate here, because there may be confusion with the knockdown of SLMAP3 using shRNA in NE-4C cells.
4. Lines 66-68, "On postnatal day (P)0, SLMAP3-null pups were stillborn, displaying signs of late fetal or neonatal lethality and developmental deformities, as reported previously". Figure 1C shows that a majority of SLMAP3-null pups have closed NT (18 in 29), are these mutants stillborn or displaying developmental deformities?
5. The authors state that SLMAP3-null embryos were shorter and that their caudal neural plate (posterior neuropore) was significantly wider. Here again, do all null mutants present these phenotypes, in particular those with neural tube closed?
6. Line 80, "a tissue morphing process" should be "a tissue morphogenetic process".
7. The authors state that no changes in YAP phosphorylation were detected. However, in Figure 2C, an increased expression of p-YAP and YAP in the "KD" condition can be seen at E11.5 whole embryo, while in Figure 2D, no changes can be seen at E12.5 brain. I find the results confusing. The authors need to compare the expression of YAP using the same tissues at different stages.
8. Lines 111-112, "They coordinate a multidirectional signal to cells simultaneously via antagonizing effects of the disheveled isoforms -1, -2, -3 (DVL1, DVL2, DVL3) and Prickle proteins...". What does this sentence mean? What are the antagonizing effects between "core" PCP proteins?
9. In Figure 3, data on the increased expression of Dvl2 and p-Cofilin and the decreased expression of Dvl3, JNK and ROCK in SLMAP3-null mutants are rather confusing. What would be the outcome of these changes on PCP signaling and cytoskeletal rearrangements?
10. Western blot shows a strongly increased expression of Dvl2 in SLMAP3 homozygous mutant (Figure 3C). However, there seems to be no difference or a weak decrease of p-DVL2 in immunofluorescence in the mutant (Figure 3D). These data are not fully consistent. The authors also need to examine p-DVL2 expression level by western blot.
11. I am not fully convinced by the results in Figure 3D. The authors state that they observed significantly reduced colocalization of Fzd6 and p-DVL2. However, this may be due to a decreased expression of both Fzd6 and p-DVL2.
12. In Figure 4G, it is not possible to see the neuroepithelium in SLMAP3 null mutant.
13. Figures are not well organized and presented.
14. I am not sure that it is necessary to present the results of NE-4C cells. Instead, the authors should focus their analysis on ABP and PCP by providing more convincing results.

Reviewer #3 (Comments to the Authors (Required)):

In this revised manuscript, the authors have addressed some of the concerns raised in my previous review. However, there are still a number of issues that need to be clarified by thorough discussion and additional analyses.

Thanks for taking the time to review the revised manuscript and provide constructive advice to improve the composition. We have attempted to address all the points and the manuscript is improved as a consequence.

1. The authors state that they used Nestin to monitor neuroepithelial cell differentiation. Is Nestin considered as a marker of differentiation? The expression of Nestin was only examined at E9.5 (Figure 4G), which seems to show a decrease or absence at the apical side. The expression of N-cadherin is associated with neuroepithelial cell differentiation, but this was also examined at a single stage (E8.5 in Figure S3A), and there are no changes in its distribution in the neural tube at this stage. Therefore, it is unclear whether the loss of SLMAP3 affects neuroepithelial cell differentiation.

Thanks for bringing these points up. You are correct, Nestin is more concentrated basally (Figure 4G). About the differentiation, we agree that N-cadherin is a more suitable marker for differentiation and we have now included a new image for N-cadherin staining at E9.5 in Figure S3A, which also shows no change in expression at this stage, implying no differentiation deficit.

2. The authors suggest that "SLMAP3 may have a mode of action in neural tube development through post-transcriptional mechanisms". How does SLMAP3 post-transcriptionally regulate neural tube development? This should be discussed by presenting arguments supporting the hypothesis.

Great point, thank you. The RNA-seq did not reveal any changes in genes commonly associated with neurulation and craniorachischisis although the neural tubes are open. Alternative mechanism operating at the protein level must be impacted by SLMAP3 loss. We replaced "through post-transcriptional mechanisms" for "at the post-translational level" for clarity.

3. In Figure 1C and D, I suppose that "KD" refers to heterozygous mutants. The use of "KD" is not appropriate here, because there may be confusion with the knockdown of SLMAP3 using shRNA in NE-4C cells.

We agree it is confusing, thank you. For clarity, we replaced KO, KD and WT in Figures 1A, 1C and 2C for "-/-, +/- and +/+". Figure 1D does not have any KD, just KO and WT, so we kept the way it is.

4. Lines 66-68, "On postnatal day (P)0, SLMAP3-null pups were stillborn, displaying signs of late fetal or neonatal lethality and developmental deformities, as reported previously". Figure 1C shows that a majority of SLMAP3-null pups have closed NT (18 in 29), are these mutants stillborn or displaying developmental deformities?

Although the animals with closed neural tube can be born alive, all SLMAP3-null embryos exhibit organ abnormalities and lethality. For clarity, we have rewrote this to "On postnatal day (P)0,

SLMAP3-null pups were stillborn, and all displayed signs of late fetal or neonatal lethality and developmental deformities, as reported previously”.

5. The authors state that SLMAP3-null embryos were shorter and that their caudal neural plate (posterior neuropore) was significantly wider. Here again, do all null mutants present these phenotypes, in particular those with neural tube closed?

Yes, all null mutants display these phenotypes. Because the measurements were done at E8.5 when the neurulation is not complete yet, we can't tell which mutants would have open or closed neural tube.

6. Line 80, "a tissue morphing process" should be "a tissue morphogenetic process".

You are correct and we have corrected this term. Thank you.

7. The authors state that no changes in YAP phosphorylation were detected. However, in Figure 2C, an increased expression of p-YAP and YAP in the "KD" condition can be seen at E11.5 whole embryo, while in Figure 2D, no changes can be seen at E12.5 brain. I find the results confusing. The authors need to compare the expression of YAP using the same tissues at different stages.

The stain free indicates that in the KD (now +/-) lane there is more protein loading, which is consistent with increased amount of p-YAP and total YAP. Further, quantification of the data with three biological replicates indicate no significant difference in p-YAP/YAP due to SLMAP3 loss. We did analyze YAP in the same tissues for WT, KO and KD at E11.5 whole embryos and at E12.5 brains. These analyses support the contention that SLMAP3 KO does not affect p-YAP and Hippo signaling. Further, this data is consistent with what we recently reported in embryonic cardiac myocytes where the specific deletion of SLMAP3 had no impact on p-YAP (PMID: 38474134).

8. Lines 111-112, "They coordinate a multidirectional signal to cells simultaneously via antagonizing effects of the disheveled isoforms -1, -2, -3 (DVL1, DVL2, DVL3) and Prickle proteins...". What does this sentence mean? What are the antagonizing effects between "core" PCP proteins?

We meant that Fzd/Dvl and Vangl/Prickle modules of the core PCP antagonize each other to allow the exclusive localization in opposite sides in the cell, creating the polarity. The mechanism for this is still not clear, but it is based on competitive interactions between PCP components that ultimately creates the asymmetry (PMID: 34896020). This information has been included in the first paragraph of the section “2.3. Loss of SLMAP3 perturbs components of planar cell polarity” for clarity.

9. In Figure 3, data on the increased expression of Dvl2 and p-Cofilin and the decreased expression of Dvl3, JNK and ROCK in SLMAP3-null mutants are rather confusing. What would be the outcome of these changes on PCP signaling and cytoskeletal rearrangements?

DVL2, DVL3, JNK, ROCK and Cofilin are part of the transducing pathway of PCP that is responsible for cytoskeleton rearrangement that dictate cell polarity and convergent extension. Disturbances in these ultimately affects PCP, which could explain the phenotype observed in

SLMAP3 neural tubes. This information is in the first paragraph of section “2.3. Loss of SLMAP3 perturbs components of planar cell polarity” and in the fifth paragraph of the Discussion section.

10. Western blot shows a strongly increased expression of Dvl2 in SLMAP3 homozygous mutant (Figure 3C). However, there seems to be no difference or a weak decrease of p-DVL2 in immunofluorescence in the mutant (Figure 3D). These data are not fully consistent. The authors also need to examine p-DVL2 expression level by western blot.

In the Western Blot, we analyzed total DVL2 in brain lysates, whereas in the immunofluorescence we assessed pDVL2 in neural tubes. Increased total DVL2 does not necessarily mean increased phosphorylation status. Finally, given the amount of protein required for Western Blot, it would be a challenge to perform Western Blot with neural tube lysates.

11. I am not fully convinced by the results in Figure 3D. The authors state that they observed significantly reduced colocalization of Fzd6 and p-DVL2. However, this may be due to a decreased expression of both Fzd6 and p-DVL2.

Decreased expression of both Fzd6 and p-DVL2 does not necessarily mean less colocalization. For this measurement, we utilized a plugin on Fiji (JaCoP) that takes into account the staining of Fzd6 and p-DVL2 for every single pixel and gives the correlation coefficient between them. Thus, both can still have less colocalization, regardless of more or less expression. The method for this measurement is detailed in the fourth paragraph of “5.2 Histological and Immunofluorescent analysis” in the section “5. Materials and Methods”.

12. In Figure 4G, it is not possible to see the neuroepithelium in SLMAP3 null mutant.

Nestin staining is more concentrated in the basal region of the neural tube of SLMAP3 null mutant embryos, that is why the entire neuroepithelium not visible, further indicating that SLMAP is required for the proper distribution of Nestin. The neuroepithelium is visible in the DNA and merge panels for SLMAP3 null mutant, and the arrowhead already indicates the apical region (Figure 4G).

13. Figures are not well organized and presented.

Based on previous reviews, we reorganized all the figures, including moving some to supplementary material as well as quantifying the images as requested. We hope that this is acceptable now.

14. I am not sure that it is necessary to present the results of NE-4C cells. Instead, the authors should focus their analysis on ABP and PCP by providing more convincing results.

Thank you for the comment, but it is prudent to show the results in these cells because they indicate no changes in the recruitment of ABP/PCP components at cell-cell contacts, implying further that *in vivo* mechanisms are often not recapitulated in culture and that SLMAP3 may operate through a signalling mechanism that is absent in cultured cells.

September 24, 2024

RE: Life Science Alliance Manuscript #LSA-2023-02545-TRR

Prof. Balwant S Tuana
University of Ottawa
Cellular and Molecular Medicine
451 Smyth Rd
Ottawa, ONT K1H8M5
Canada

Dear Dr. Tuana,

Thank you for submitting your revised manuscript entitled "SLMAP3 is essential for neurulation through mechanisms involving cytoskeletal elements, ABP and PCP". We would be happy to publish your paper in Life Science Alliance pending final revisions necessary to meet our formatting guidelines.

- please be sure that the authorship listing and order is correct
- please upload your table files as editable doc or excel files
- please add the Twitter handle of your host institute/organization as well as your own or/and one of the authors in our system
- please consult our manuscript preparation guidelines <https://www.life-science-alliance.org/manuscript-prep> and make sure your manuscript sections are in the correct order and please incorporate your conclusion section into your Discussion section
- please use the [10 author names, et al.] format in your references (i.e. limit the author names to the first 10)
- please add a callout for your Figure S5A; you have a callout for Figure S5C but this is not in the figure or the figure legend for S5-please correct
- you may want to consider uploading Figure 8 as a Graphical Abstract rather than as a figure, but this it up to you

A. FINAL FILES:

B. MANUSCRIPT ORGANIZATION AND FORMATTING:

Thank you for your attention to these final processing requirements. Please revise and format the manuscript and upload materials within 5 days.

Sincerely,

September 26, 2024

RE: Life Science Alliance Manuscript #LSA-2023-02545-TRRR

Prof. Balwant S Tuana
University of Ottawa
Cellular and Molecular Medicine
451 Smyth Rd
Ottawa, ONT K1H8M5
Canada

Dear Dr. Tuana,

Thank you for submitting your Research Article entitled "SLMAP3 is essential for neurulation through mechanisms involving cytoskeletal elements, ABP and PCP". It is a pleasure to let you know that your manuscript is now accepted for publication in Life Science Alliance. Congratulations on this interesting work.

DISTRIBUTION OF MATERIALS:

Again, congratulations on a very nice paper. I hope you found the review process to be constructive and are pleased with how the manuscript was handled editorially. We look forward to future exciting submissions from your lab.

Sincerely,
